# DOT1L safeguards cartilage homeostasis and protects against osteoarthritis

Silvia Monteagudo[1], Frederique M.F. Cornelis[1], Carolina Aznar-Lopez[1], Ploi Yibmantasiri[1], Laura-An Guns[1], Peter Carmeliet[2,3], Frédéric Cailotto[1,4] & Rik J. Lories[1,5]

Osteoarthritis is the most prevalent and crippling joint disease, and lacks curative treatment, as the underlying molecular basis is unclear. Here, we show that DOT1L, an enzyme involved in histone methylation, is a master protector of cartilage health. Loss of DOT1L disrupts the molecular signature of healthy chondrocytes *in vitro* and causes osteoarthritis in mice. Mechanistically, the protective function of DOT1L is attributable to inhibition of Wnt signalling, a pathway that when hyper-activated can lead to joint disease. Unexpectedly, DOT1L suppresses Wnt signalling by inhibiting the activity of sirtuin-1 (SIRT1), an important regulator of gene transcription. Inhibition of SIRT1 protects against osteoarthritis triggered by loss of DOT1L activity. Modulating the DOT1L network might therefore be a therapeutic approach to protect the cartilage against osteoarthritis.

[1] Laboratory of Tissue Homeostasis and Disease, Skeletal Biology and Engineering Research Center, Department of Development and Regeneration, KU Leuven, Leuven 3000, Belgium. [2] Laboratory of Angiogenesis and Vascular Metabolism, Department of Oncology, KU Leuven, Leuven 3000, Belgium. [3] Laboratory of Angiogenesis and Vascular Metabolism, Vesalius Research Center, VIB, Leuven 3000, Belgium. [4] CNRS-Université de Lorraine, UMR7365, Ingénierie Moléculaire et Physiopathologie Articulaire (IMoPA), Biopôle de l'Université de Lorraine, Campus Biologie-Santé, Vandœuvre-Lès-Nancy 54500, France. [5] Division of Rheumatology, University Hospitals Leuven 3000, Belgium. Correspondence and requests for materials should be addressed to R.J.L. (email: Rik.Lories@kuleuven.be).

Articular cartilage is an essential component of our joints and critical for normal mobility. This cartilage contains a unique cell-type, the articular chondrocyte, which is embedded in self-produced extracellular matrix composed mainly of type 2 collagen fibres and proteoglycan aggrecan. Osteoarthritis, the most common joint disease, is characterized by progressive damage to the articular cartilage. In osteoarthritis, the chondrocytes die or lose their highly specialized molecular characteristics, resulting in the production of an extracellular matrix that is biomechanically inferior, contributing to progressive tissue damage and loss of joint function[1]. Thus, this age-related or trauma-triggered progressive disease poses a medical health threat and debilitates affected patients. Current therapy is limited to symptom relief and in severe cases joint replacement surgery; interventions that arrest or reverse disease progression are entirely lacking and therefore indicate a large unmet medical need[2]. Effective anti-osteoarthritic drugs should maintain cartilage homeostasis and structural integrity, but the central molecular regulators of these processes are unknown, precluding the development of effective, safe treatments.

The Disruptor of telomeric silencing 1-like (DOT1L) gene encodes a histone methyltransferase that methylates lysine-79 of histone H3 (H3K79) and is involved in epigenetic regulation of gene transcription[3–5]. Genome-wide association studies showed that common single variants in the DOT1L gene, with minor allele frequencies of 0.20–0.40, protect against osteoarthritis[6,7]. However, it is unknown how DOT1L affects this devastating disease.

In this study, we identify DOT1L as a regulator of cartilage health and disease. Loss of DOT1L disrupts cartilage homeostasis and triggers the development of osteoarthritis. Furthermore, DOT1L preserves cartilage health by preventing the hyper-activation of Wnt signalling through negative regulation of SIRT1. Overall, our data demonstrate the importance of the DOT1L/SIRT1 axis in maintaining cartilage health and provides a rationale for potential therapeutic interventions in the treatment of osteoarthritis.

## Results

**Loss of DOT1L activity disrupts chondrocyte homeostasis.** To study whether cartilage degeneration in osteoarthritis is related to changes in DOT1L activity, we performed immunohistochemistry of DOT1L-methylated H3K79 on cartilage from non-osteoarthritic trauma patients and on preserved and damaged regions of cartilage from patients with osteoarthritis (Fig. 1a). This analysis revealed that the immunoreactive signal of methylated H3K79 was decreased in damaged areas from patients with osteoarthritis as compared to their corresponding preserved areas and to control cartilage. In contrast, DOT1L gene expression did not differ between damaged or preserved cartilage from patients with osteoarthritis (average fold change in damaged versus preserved cartilage 0.992 (s.e.m. 0.064), $n = 4$ patients, with three technical replicates). These observations suggest that DOT1L activity positively correlates with cartilage health.

We next sought to determine if DOT1L regulates transcriptional programs that maintain the unique molecular signature of the articular chondrocyte. To address this question, we used an in vitro model in which freshly isolated healthy human articular chondrocytes are cultured and serially passaged in monolayer, thereby losing their molecular characteristics, as it occurs in osteoarthritis[8]. In this osteoarthritis-like dedifferentiation process, the expression of chondrocyte markers such as type 2 collagen (COL2A1) and aggrecan (ACAN) is lost, while fibroblast markers such as type 1 collagen (COL1A1) are upregulated[9]. We

examined how DOT1L inhibition affects gene expression changes triggered in this model. When the articular chondrocytes were expanded and passaged while treated with the specific DOT1L inhibitor EPZ-5676, H3K79 methylation was successfully abrogated (Supplementary Fig. 1a,b). DOT1L inhibition enhanced the osteoarthritis-like gene expression changes, for example, augmenting loss of COL2A1 and gain of COL1A1 expression (Fig. 1b). These results suggest that the methyltransferase plays a role in maintaining the molecular identity of the healthy articular chondrocyte.

**In vivo loss of DOT1L activity triggers osteoarthritis.** We then studied whether loss of DOT1L activity in vivo triggers osteoarthritis, by intra-articular injection of EPZ-5676 into the knees of adult mice. H3K79 methylation was effectively inhibited by EPZ-5676 in articular chondrocytes (Fig. 1c). We observed increased cartilage damage by histology 2 and 4 weeks after EPZ-5676 injections (Fig. 1d,e). There were no differences between the groups in severity of synovitis or extent of subchondral bone remodelling, tissues that are known to show other features of osteoarthritis[10] (Supplementary Fig. 2). Thus, these data suggest that DOT1L preserves articular cartilage homeostasis, protecting it against osteoarthritis.

**DOT1L limits Wnt signalling to maintain cartilage homeostasis.** To understand the mechanism by which DOT1L controls cartilage homeostasis and to identify effector signalling pathways, we performed genome-wide transcriptome analysis in healthy human articular chondrocytes treated with EPZ-5676 or vehicle (geonr: GSE77916). Microarray results and subsequent qPCR validation confirmed the induction of osteoarthritis-like gene expression changes upon DOT1L blockade (Supplementary Fig. 3a and Supplementary Table 1). Comparative pathway analyses of differentially regulated transcripts after EPZ-5676 treatment revealed an enrichment in genes associated with Wnt signalling (Fig. 2a). Additional analyses highlighted the diverse regulatory role of DOT1L and further supported the links with skeletal biology and disease (Supplementary Fig. 3b,c).

Tight regulation of Wnt signalling is key to cartilage health as both insufficient and excessive high levels have been associated with osteoarthritis[11–13]. Upon Wnt ligand binding to Fizzled receptors and co-receptors including low-density lipoprotein receptor-related protein (LRP) 5/6, β-catenin is rescued from proteasomal degradation, accumulates in the cytoplasm and translocates into the nucleus, where it associates with transcription factors from the TCF/LEF family. Immunoprecipitation analysis demonstrated that DOT1L interacts with β-catenin, the key signalling molecule in the Wnt cascade (Fig. 2b).

To find out how DOT1L affects Wnt signalling, we determined the effects of DOT1L inhibition on the activation of this pathway in healthy human articular chondrocytes and in mouse models. In Wnt reporter-transfected cells, EPZ-5676 increased the luciferase activity when the Wnt signalling pathway was activated by LiCl (Fig. 2c). However, DOT1L inhibition did not induce detectable changes in active β-catenin levels in human articular chondrocytes (Fig. 2b) or in the articular cartilage of EPZ-5676-injected mice (Supplementary Fig. 4a). This suggests that DOT1L regulates Wnt signalling downstream of β-catenin stabilization. Effectively, messenger RNA levels of direct Wnt target genes (LEF1, TCF1 and c-MYC)[14–16] increased after DOT1L inhibition in LiCl-treated chondrocytes (Fig. 2d). Likewise, DOT1L knockdown using small interfering RNA (siRNA) led to upregulation of these Wnt target genes in LiCl-treated cells (Fig. 2e). In another set-up of Wnt signalling

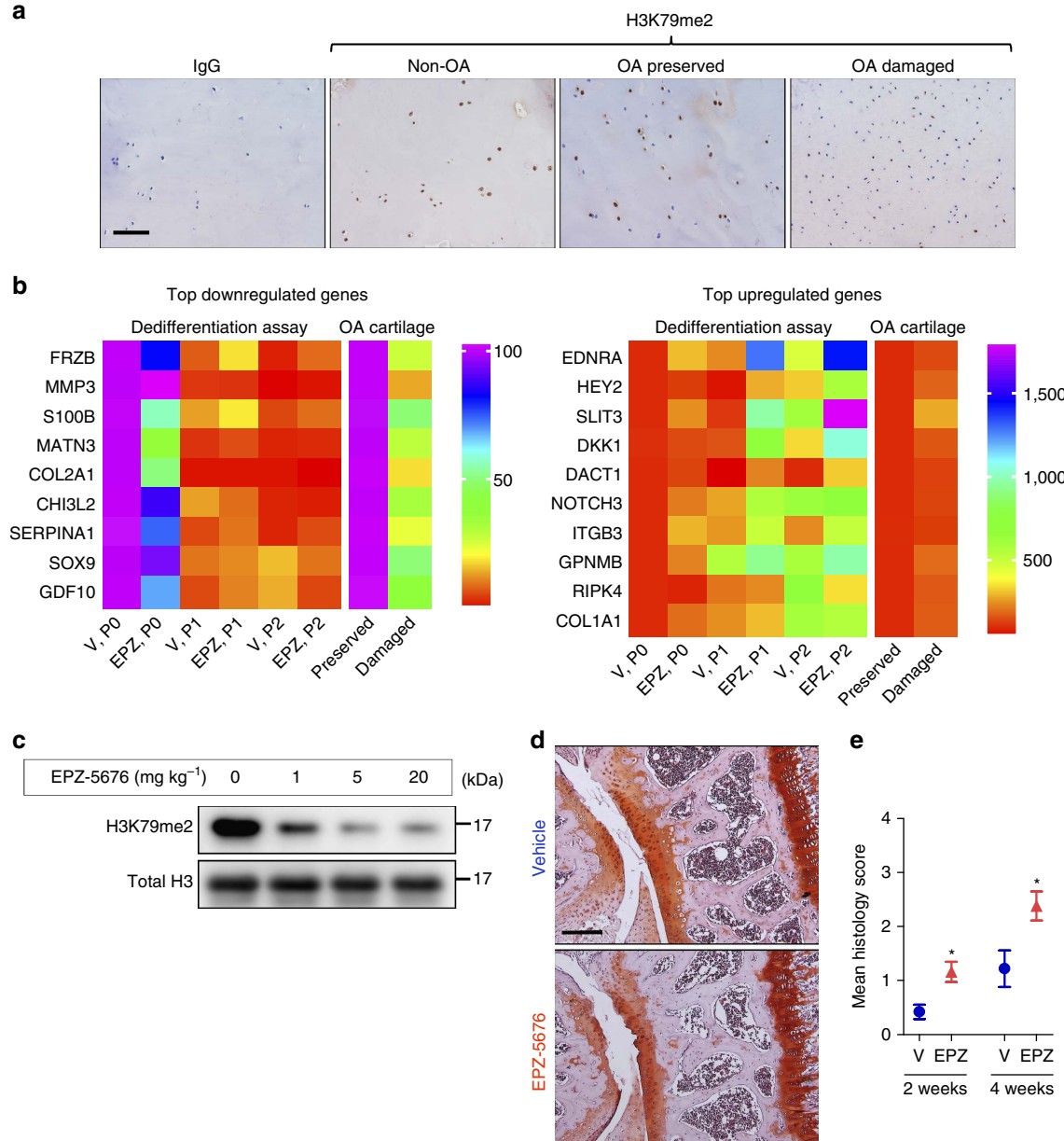

**Figure 1 | Loss of DOT1L disrupts chondrocyte homeostasis and triggers osteoarthritis.** (**a**) Immunohistochemistry showing reduced methylated H3K79 levels (H3K79me2) that reflect loss of DOT1L activity in damaged areas from osteoarthritic patients (OA) as compared to their corresponding preserved areas and to cartilage from non-OA patients. Images are representative of images from four different patients. Scale bar, 400 µm. (**b**) Heat maps of differential mRNA expression determined by quantitative PCR in chondrocytes treated with DOT1L inhibitor EPZ-5676 (EPZ) or vehicle (V) from passage 0 (P0) until P2, and from preserved versus damaged areas in OA cartilage. The colour code represents the mean expression level of six and four independent patient samples respectively. (**c**) Immunoblot analysis showing decreased methylated H3K79 levels in mouse articular chondrocytes after intra-articular injection of EPZ into C57Bl/6 wild-type mouse knees. The image is representative of one experiment with protein extracts pooled from two or three mice per condition. Unprocessed original scans of blots are shown in Supplementary Fig. 10. (**d,e**) C57/Bl6 wild-type mouse knees were injected with EPZ (5 mg kg$^{-1}$) or vehicle and killed after 2 or 4 weeks. Knees were sectioned and stained with Hematoxylin-Safranin O (**d**). Scale bar, 200 µm. Cartilage damage was scored (see Methods section) and is shown in (**e**). One experiment was performed with $n = 10$ and 5. Representative images from the 4 week evaluation are shown. *$P < 0.05$ (two-tailed $t$-test). Error bars indicate mean ± s.e.m.

pathway activation using recombinant Wnt3a protein, DOT1L inhibition also resulted in increased Wnt target gene expression (Supplementary Fig. 5a). EPZ-5676 also augmented levels of these Wnt targets in the dedifferentiation assay (Supplementary Fig. 1c). *In vivo*, increased Wnt target gene expression was demonstrated by immunohistochemistry of TCF1 in adult mouse articular cartilage after DOT1L inhibition (Fig. 2f). Altogether, these data suggest that DOT1L restrains active Wnt signalling in cartilage.

To show that hyper-activation of Wnt signalling is the main mechanism responsible for the pathological effects of DOT1L loss of activity in articular cartilage, we assessed whether the Wnt inhibitor XAV-939 counteracts the effects of EPZ-5676 *in vitro* and *in vivo*. In healthy human articular chondrocytes, XAV-939 treatment rescued the loss of *ACAN*, *COL2A1* and the increase of *COL1A1* triggered by DOT1L inhibition (Fig. 3a). Likewise, *in vivo*, Wnt inhibition by XAV-939 also protected mice from EPZ-5676-induced osteoarthritic changes (Fig. 3b,c). Thus,

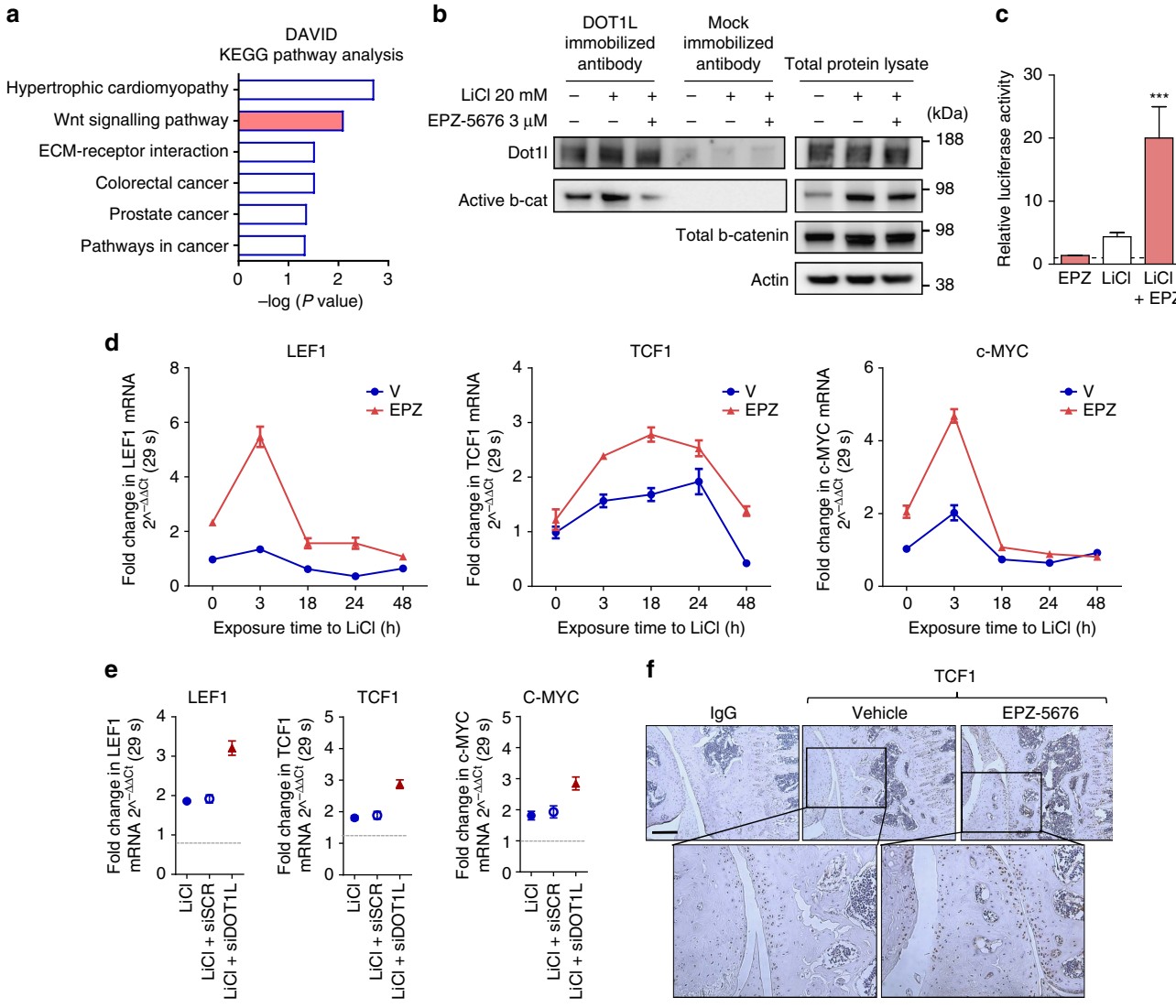

**Figure 2 | DOT1L negatively regulates Wnt signalling in articular cartilage.** (**a**) KEGG pathway enrichment analysis of microarray data obtained from human articular chondrocytes treated with EPZ-5676 or vehicle. Nominal *P* values by EASE modified Fisher Exact test using the DAVID analysis tool (see Methods section) are shown. $n = 5$ independent patient-derived cell cultures. (**b**) Co-immunoprecipitation (Co-IP) using an anti-DOT1L antibody showing interaction between DOT1L and β-catenin in human articular chondrocytes, that is increased upon Wnt activation by LiCl and disrupted upon DOT1L inhibition. The image is representative of three experiments. (**c**) TOP/FOP reporter assay in human articular chondrocytes after Wnt stimulation by LiCl and DOT1L inhibition by EPZ. Activity is compared to untreated cells (dotted line). $n = 3$ biologically independent experiments. \*\*\*$P < 0.001$ by one-way ANOVA. (**d,e**) *LEF1*, *TCF1* and *c-MYC* expression measured by quantitative PCR in chondrocytes treated with EPZ-5676 and LiCl (**d**) or in LiCl-treated chondrocytes transfected with siRNA directed against DOT1L or scrambled siRNA (siDOT1L or siSCR, respectively) (**e**). Data are from one experiment with three technical replicates. (**f**) Immunohistochemistry demonstrating increased TCF1 levels in the articular cartilage of C57/Bl6 wild-type mice after injection of EPZ-5676. The images are representative of three different animals. Scale bar, 200 μm.

hyper-activation of Wnt signalling is the major deleterious downstream effect of loss of DOT1L activity in cartilage.

Chondrocytes are also found in the growth plate cartilage, a transient tissue that becomes gradually replaced by bone during skeletal development and growth. In developmental bone formation and in the growth plate, active Wnt signalling has an important role in terminal differentiation of these chondrocytes towards hypertrophic cells. These cells express type X collagen (COLX), upregulate matrix metalloproteinase-13 (MMP-13) and produce a calcified extracellular matrix. In osteoarthritis, hyper-activation of Wnt signalling is associated with ectopic hypertrophic differentiation[11,17]. Altered matrix composition and factors secreted by these hypertrophic-like articular chondrocytes, such as MMP-13, likely contribute to cartilage degeneration in osteoarthritis[17]. We detected increased immunohistochemical

staining of COLX and MMP-13 in the articular cartilage of EPZ-5676-injected mice, particularly in the vicinity of lesions (Supplementary Fig. 4b,c). Thus, DOT1L also protects against osteoarthritis by preventing Wnt-associated ectopic chondrocyte hypertrophy in the articular cartilage.

**DOT1L controls Wnt activity by negative regulation of SIRT1.** To investigate whether DOT1L directly regulates the Wnt pathway at the transcriptional level, we studied the binding of DOT1L to and its activity on different Wnt target genes in human articular chondrocytes. Chromatin immunoprecipitation-quantitative PCR (ChIP-qPCR) demonstrated that DOT1L and methylated H3K79 bound to *LEF1* and *TCF1,* but not to *c-MYC* genes around the transcriptional start site (TSS) in response to

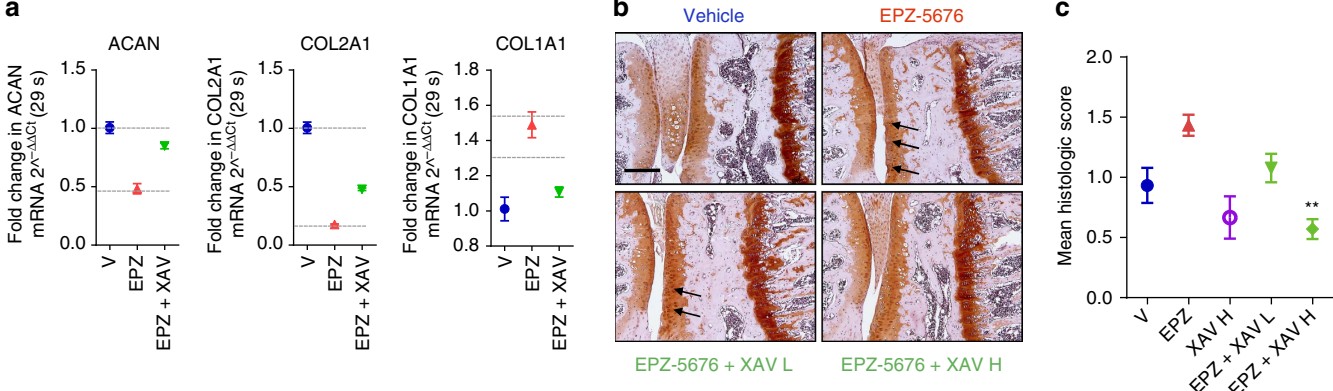

**Figure 3 | DOT1L protects against osteoarthritis by preventing Wnt hyper-activation.** (**a**) Wnt antagonist XAV-939 (XAV) rescues gene expression changes in *ACAN*, *COL2A1* and *COL1A1* induced by EPZ-5676 in healthy human articular chondrocytes. Data are from one experiment with three technical replicates. (**b**,**c**) C57/Bl6 wild-type mouse knees were injected with EPZ (5 mg kg$^{-1}$), XAV (0.1 (low dose—L) or 0.5 (high dose—H) mg kg$^{-1}$) or vehicle and killed after 4 weeks. Knees were sectioned and stained with Hematoxylin-Safranin O (**b**). Scale bar, 200 μm. Cartilage damage was scored (see Methods section) and is shown in (**c**). One experiment was performed with $n = 3$ (vehicle, EPZ and XAV-939 only) and $n = 10$ (EPZ + XAV9393 groups). **$P < 0.01$ (two-tailed *t*-test) compared to EPZ treatment only. All error bars indicate mean ± s.e.m.

Wnt pathway activation with either LiCl or Wnt3a (Fig. 4a and Supplementary Fig. 5b). The binding strongly decreased upon EPZ-5676 treatment or siRNA-mediated DOT1L knockdown (Fig. 4a and Supplementary Fig. 5b). Loss of H3K79 methylation was accompanied by higher levels of acetylated H3K9 and trimethylated H3K4, histone marks associated with active transcription, at the *LEF1* and *TCF1*, but not *c-MYC* promoters (Fig. 4b). Thus, DOT1L acts as a direct negative regulator of Wnt target genes in chondrocytes.

We next sought to elucidate the molecular mechanism by which DOT1L negatively regulates Wnt signalling in cartilage. We hypothesized that DOT1L interacts with a repressor to restrict Wnt target gene expression. To test our hypothesis, we assessed whether knockdown of potential repressor candidates mimicked the effects of DOT1L inhibition in LiCl-treated human articular chondrocytes. As candidates, we evaluated AF4, AF5 and AF10, which are known DOT1L binding partners in leukaemia[18–20] with reported roles in gene repression[21,22] or as enhancers of DOT1L effects[23]. We also considered as candidates BCOR, SIRT1 and CBX8, proteins earlier identified in DOT1L complexes by mass spectrometry, with a known role as transcriptional repressors[24]. Silencing of any of these candidates did not upregulate *LEF1* or *TCF1* expression in Wnt-activated cells (Fig. 4c and Supplementary Fig. 6a). Therefore, we did not further consider their possible role as transcriptional repressors in interaction with DOT1L.

Surprisingly, in DOT1L-inhibited cells, we found that silencing of *SIRT1* blocked the upregulation of *LEF1*, *TCF1* but not *c-MYC* observed after DOT1L inhibition (Fig. 4c and Supplementary Fig. 6a). Likewise, the SIRT1 specific inhibitor EX527 rescued the upregulation of *LEF1* and *TCF1* triggered by DOT1L inhibition in LiCl-treated cells (Fig. 4d and Supplementary Fig. 6b). In contrast, pharmacological activation of SIRT1 with SRT1720 in LiCl/EPZ-5676-treated cells further upregulated *LEF1*, *TCF1* but not *c-MYC* (Fig. 4d and Supplementary Fig. 6b). These unexpected results prompted us to consider that SIRT1 drives Wnt hyper-activation upon DOT1L blockade in chondrocytes.

We next investigated the mechanism by which SIRT1 increases Wnt target gene expression upon DOT1L inhibition. Immunoprecipitation demonstrated that DOT1L and SIRT1 interacted at the protein level in human articular chondrocytes and this interaction was disrupted by DOT1L inhibition (Fig. 4e). Notably, we found that DOT1L inhibition increased the enzymatic activity of SIRT1 (Fig. 4f). DOT1L knockdown using siRNA likewise resulted in enhanced SIRT1 activity (Fig. 4f). However, since ChIP-qPCR showed no SIRT1 protein occupancy at the DOT1L-identified Wnt target genes upon DOT1L inhibition in our experimental conditions (Fig. 4g and Supplementary Fig. 6c), we hypothesized that SIRT1's effects may be accomplished by downstream chromatin binding factors. We therefore focused on PPARGC1A, GCN5 and EP300, transcriptional activators linked to the SIRT1 network[25]. Silencing of these factors mimicked SIRT1 silencing and partially blocked *LEF1* and *TCF1* but not *c-MYC* upregulation by EPZ-5676 in LiCl-treated chondrocytes (Fig. 4h and Supplementary Fig. 6d). ChIP-qPCR demonstrated strong enrichment for PPARGC1A and to a lesser extent for GCN5 at the *LEF1* and *TCF1* promoters after EPZ-5676 treatment (Fig. 4g and Supplementary Fig. 6c). Furthermore, the SIRT1 inhibitor EX527 that rescued EPZ-5676 effects on Wnt targets (Fig. 4d and Supplementary Fig. 6b), reduced the occupancy of PPARGC1A and GCN5 at *LEF1* and *TCF1* promoters (Fig. 4g and Supplementary Fig. 6c). Thus, these results indicate that increased SIRT1 activity upon DOT1L inactivation induces chromatin binding of transcriptional activators and promotes the upregulation of Wnt target genes.

**SIRT1 blockade prevents EPZ-5676-induced osteoarthritis**. We then examined the therapeutic implications of our findings, and explored the therapeutic potential of blocking SIRT1 in preventing osteoarthritis triggered by DOT1L inactivation. We therefore intra-articularly co-injected the SIRT1 inhibitor EX527 and the DOT1L inhibitor EPZ-5676. Pharmacological blockade of SIRT1 effectively reduced the severity of osteoarthritis caused by the DOT1L inhibitor (Fig. 5a,b), and reversed the hyper-activation of Wnt signalling (Fig. 5c). Thus, we discovered that modulation of SIRT1 activity is a leading mechanism by which DOT1L controls activation of Wnt signalling, maintains cartilage health and prevents osteoarthritis.

**Loss of DOT1L causes severe growth retardation in mice**. The DOT1L inhibitor EPZ-5676 is currently under clinical investigation as a potential treatment for MLL-rearranged leukaemia in adults and children[26,27]. Our loss of function data in the articular cartilage resulting in osteoarthritis suggest that

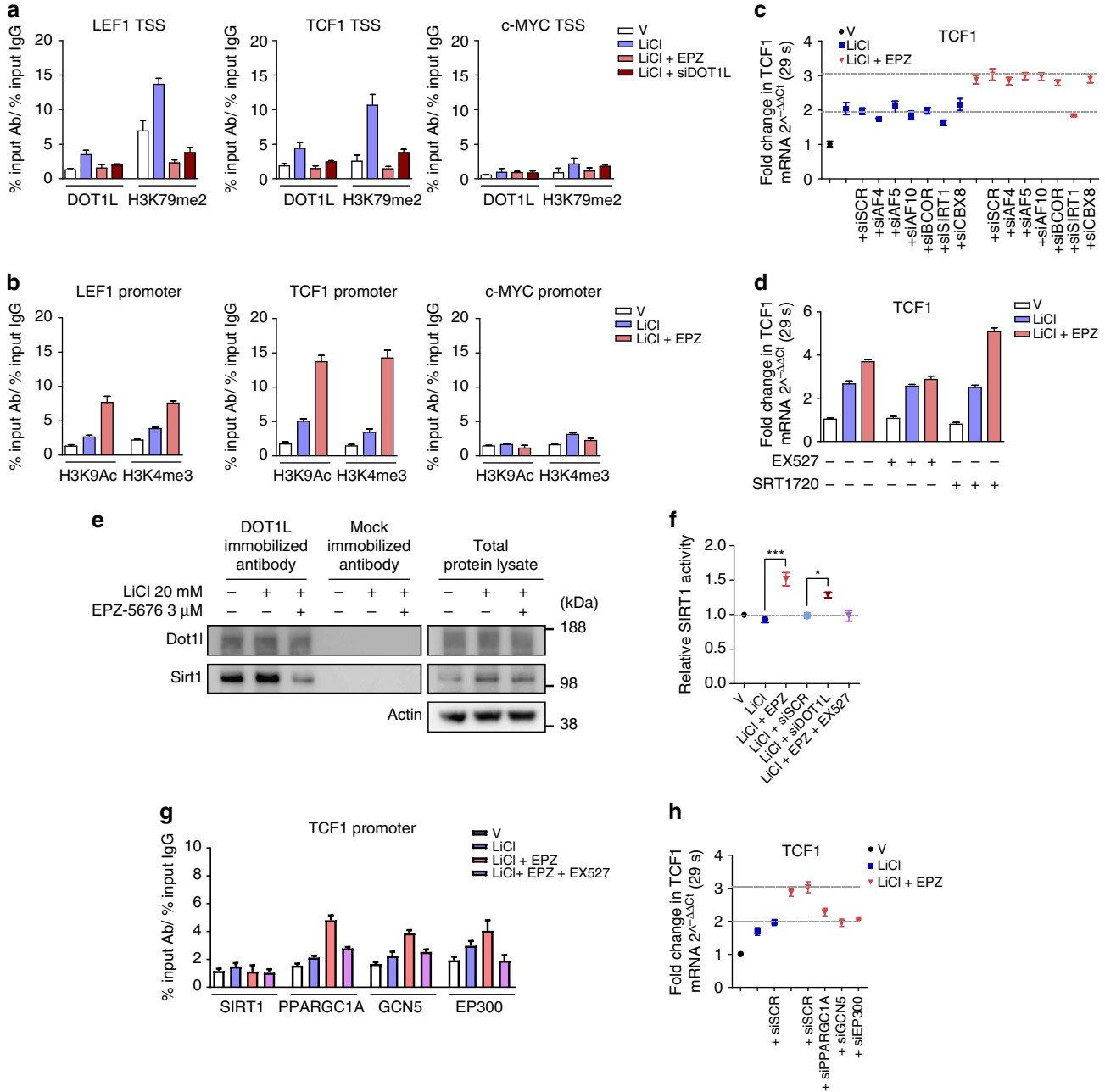

**Figure 4 | DOT1L directly controls Wnt targets by negative regulation of SIRT1.** All experiments were performed in healthy human articular chondrocytes: treated as indicated with DOT1L inhibitor EPZ-5676, Wnt activator LiCl, SIRT1 antagonist EX527 or SIRT1 agonist SRT1720; or transfected with DOT1L or scrambled siRNA. All data are presented as mean ± s.e.m. (**a**) Chromatin immunoprecipitation-quantitative PCR (ChIP-qPCR) analysis of DOT1L and methylated H3K79 and (**b**) acetylated H3K9 (H3K9Ac) and methylated H3K4 (H3K4me3) as markers of active transcription on the transcriptional start site (TSS) of Wnt target genes. Data are from two to five experiments. (**c**) Expression levels of *TCF1* Wnt target gene measured by quantitative PCR in chondrocytes transfected with indicated specific or scrambled siRNA (siSCR). Data are from one experiment with technical triplicates. (**d**) *TCF1* expression measured by quantitative PCR in the presence of SIRT1 agonist and antagonist. Data from two experiments each with technical triplicates. (**e**) Co-IP analysis using the indicated antibodies demonstrating the interaction of DOT1L and SIRT1. The image is a representative image of three biologically independent experiments. (**f**) SIRT1 activity relative to vehicle-treated cells (dotted line). Data are from three biologically independent experiments. *$P < 0.05$, ***$P < 0.001$ by one-way ANOVA. (**g**) ChIP-qPCR analysis of SIRT1, PPARGC1A, GCN5 and EP300 binding on the *TCF1* promoter and (**h**) *TCF1* expression after siRNA transfection with indicated specific or scrambled siRNA. Data from two biologically independent experiments.

specific attention should be given to cartilage in patients treated with DOT1L inhibitors. Of note, polymorphisms in the *DOT1L* gene are not only associated with osteoarthritis but also with height[28]. To assess whether loss of DOT1L has also an impact on growth plate chondrocytes and may influence growth in children treated with DOT1L inhibitors, we bred floxed *Dot1l* mice

(*Dot1l$^{fl/fl}$*) with a *Col2-cre* deleter mouse strain to obtain specific recombination in chondrocytes (Supplementary Fig. 7). Cartilage-specific *Dot1l* knockout mice (*Dot1l$^{Cart-KO}$* mice) did not show apparent skeletal abnormalities but displayed severe growth retardation (Fig. 5d). Histology of the growth plates demonstrated a reduced and disorganized proliferative and

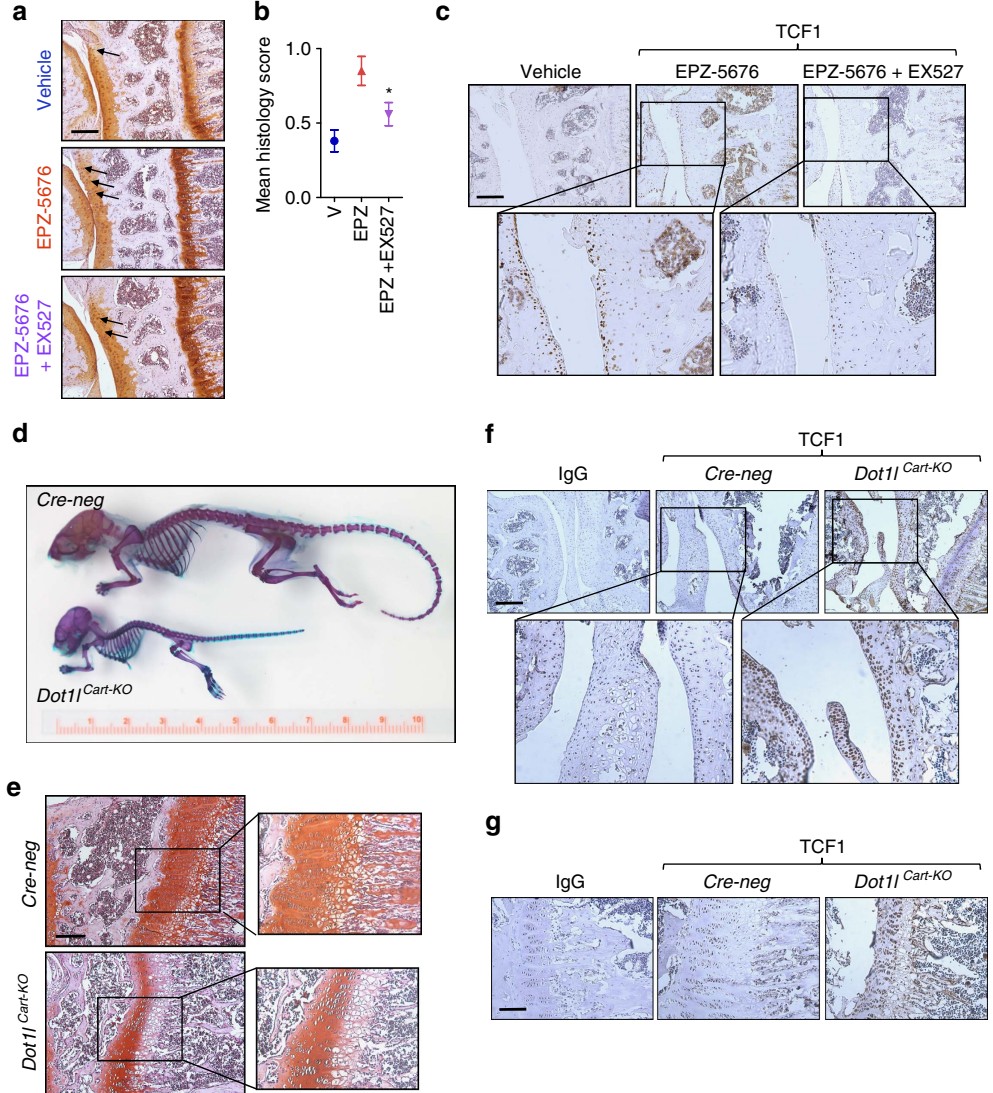

**Figure 5 | Clinical implications of modulating the DOT1L network in cartilage.** (**a–c**) Inactivation of SIRT1 protects against DOT1L inhibitor-induced osteoarthritis: (**a**) C57/Bl6 wild-type mouse knees were injected with DOT1L inhibitor EPZ-5676 (5 mg kg$^{-1}$) and SIRT1 inhibitor EX527 (1.25 mg kg$^{-1}$), or vehicle (V) and killed after 4 weeks. Knees were sectioned and stained with Hematoxylin-Safranin O (**a**). Scale bar, 200 μm. Cartilage damage was scored (see Methods section) and is shown in (**b**). One experiment was performed with $n = 3$ (vehicle), 8 (EPZ) and 10 (EPZ + EX527). *$P < 0.05$ by one-way ANOVA. Error bars indicate mean ± s.e.m. (**c**) Immunohistochemistry of TCF1 in the indicated groups. TCF1 levels are increased after EPZ treatment and normalized by additional EX527 treatment. The images are representative of three different animals. Scale bar, 200 μm. (**d,e**) Loss of DOT1L function causes severe growth retardation as demonstrated by skeletal staining (**d**) and histology of the growth plate (**e**) of 4-week-old $Dot1l^{fl/fl};Col2\text{-}Cre^{-/-}$ (Cre-neg) and $Dot1l^{fl/fl};Col2\text{-}Cre^{+/-}$ ($Dot1l^{Cart\text{-}KO}$) mice. (**f,g**) Increased TCF1 levels in $Dot1l^{Cart\text{-}KO}$ mice as shown by immunohistochemistry in the indicated mice strains in the articular cartilage (**f**) and growth plate (**g**). The images are representative of three different animals. Scale bar, 200 and 100 μm.

prehypertrophic zone (Fig. 5e). Increased Wnt pathway activation in the absence of DOT1L was demonstrated by immunohistochemistry of TCF1 in the articular cartilage and growth plate of $Dot1l^{Cart\text{-}KO}$ mice (Fig. 5f,g), with no detectable changes in β-catenin activation (Supplementary Fig. 8a,b). The pattern and the intensity of the immunoreactive signal for COLX was different in the growth plate of $Dot1l^{Cart\text{-}KO}$ mice compared to controls (Supplementary Fig. 8d). Of note, COLX levels also appeared to be increased in the articular cartilage of $Dot1l^{Cart\text{-}KO}$ mice (Supplementary Fig. 8c). Similarly, MMP-13 levels appeared to be increased in the articular cartilage of $Dot1l^{Cart\text{-}KO}$ mice (Supplementary Fig. 8e) but not in the growth plate (Supplementary Fig. 8f). Hence, these *in vivo* data demonstrate that DOT1L not only regulates cartilage homeostasis but also

skeletal growth, and caution against undesired growth effects when DOT1L inhibitors are used in children.

## Discussion

In this study, we show that DOT1L safeguards the homeostasis of the articular cartilage and protects against osteoarthritis. We propose a new signalling model in cartilage (Fig. 6) linking Wnt activation with recruitment of DOT1L multi-protein complexes to Wnt target genes such as *TCF1* and *LEF1*. Under normal conditions, DOT1L prevents Wnt hyper-activation by negative modulation of SIRT1, thereby maintaining cartilage homeostasis. Upon DOT1L loss of function, DOT1L complexes disassemble, SIRT1 activity increases and transcriptional activators are

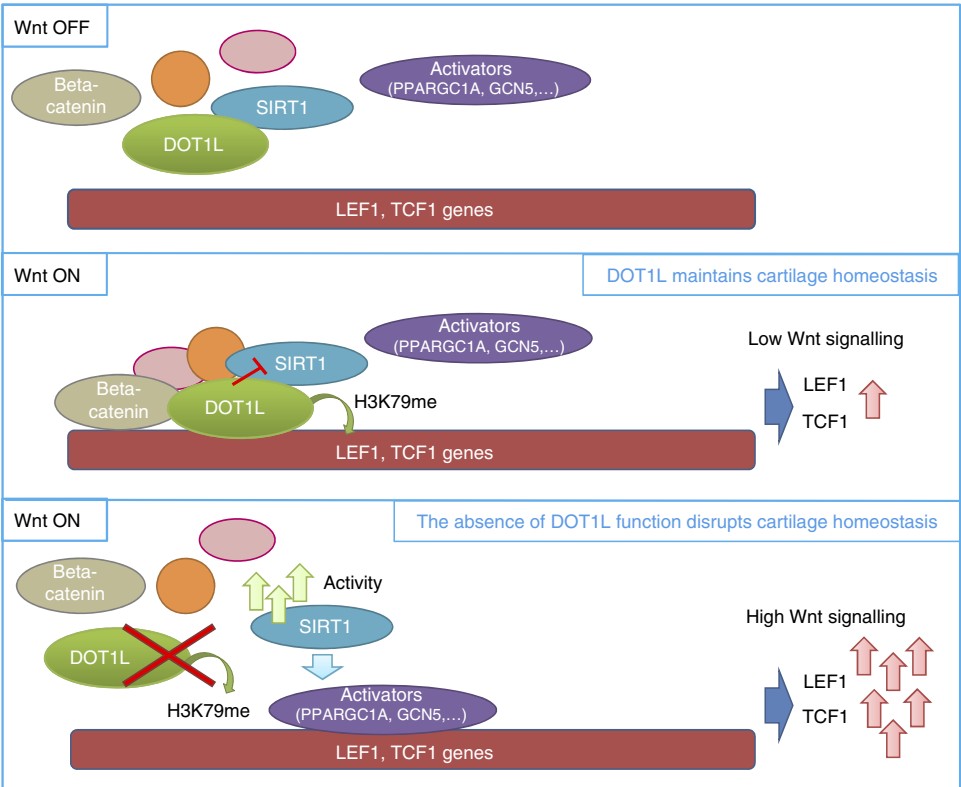

**Figure 6 | Model for the role of DOT1L in cartilage.** Upon Wnt signalling activation, DOT1L-containing complexes bind Wnt target gene chromatin. DOT1L interacts with SIRT1 and inhibits its function, preventing Wnt pathway hyper-activation. When Wnt signalling is activated in the absence of DOT1L function, high SIRT1 activity mediates the recruitment of transcriptional activators to *LEF1* and *TCF1* genes. High Wnt signalling leads to deleterious downstream effects and loss of cartilage homeostasis.

recruited in a SIRT1-dependent manner. Wnt target gene expression strongly increases, and triggers loss of cartilage homeostasis, *in vivo* resulting in osteoarthritis. Interestingly, in the joint, the effects of DOT1L on Wnt signalling appear to be cell-specific for chondrocytes as these were absent in osteoblasts (Supplementary Fig. 9).

In genome-wide association studies, polymorphisms in the *DOT1L* gene were linked with cartilage thickness and osteoarthritis[6,29,30], as well as with height[28]. We earlier demonstrated that silencing of *Dot1l* in mouse chondroprogenitor cells negatively influences chondrogenic differentiation and that DOT1L interacts with Wnt pathway transcription factor TCF4 in these cells[6]. An interaction between Wnt signalling molecule β-catenin and DOT1L was further demonstrated in human cell lines[18,31], in mouse intestinal crypts[18], in colorectal cancer[18], in drosophila[31] and in zebrafish[18]. Remarkably, and in sharp contrast with our observations in the articular chondrocytes, DOT1L and H3K79 methylation have been mostly linked to increased Wnt target gene expression[6,18,31]. Our observation that DOT1L prevents the deleterious hyper-activation of Wnt signalling in articular chondrocytes highlights that the regulatory effects of DOT1L on gene expression are strongly tissue and context dependent[32], thereby opening opportunities for highly specific interventions.

A fine-tuned balance of Wnt activity is considered indispensable for cartilage and joint homeostasis[13]. Both excessive activation and lack of Wnt signalling in the joint result in cartilage breakdown and osteoarthritis in animal models[11,33]. Our new paradigm proposes that DOT1L can act as a brake to contain the hyper-activation of the Wnt signalling cascade but does not fully inhibit its physiological role. A similar

function has recently been described for WNT16. Nalesso *et al.*[34] suggest that WNT16 prevents excessive Wnt activation in articular cartilage by acting as a partial agonist of this cascade, thereby competing for receptor binding with a full agonist. We also earlier demonstrated that lack of extracellular Wnt antagonist frizzled-related protein increases the severity of osteoarthritis in mouse models[12]. Thus, our novel data on DOT1L as a critical modulator of the Wnt cascade add another level of complexity to the regulation of Wnt signalling in the articular cartilage: in addition to extracellular agonists and antagonists, the Wnt cascade is also tightly controlled at the chromatin level. The cell-specificity of our findings suggests that this epigenetic mechanism may be a more precise therapeutic target as compared to extracellular modulators.

The factors that regulate DOT1L activity in the articular cartilage remain unknown and are an important area for further research. We did not detect differences in *DOT1L* gene expression levels between damaged or preserved cartilage from patients with osteoarthritis. However, this does not exclude transcriptional control as a mechanism to regulate DOT1L activity. In osteoarthritis, articular chondrocytes are not a uniform population and their health status may determine the expression of *DOT1L* at the individual cell level. Interestingly, some evidence suggests that pro-inflammatory signals activating NFkB signalling increase *DOT1L* expression[35]. This may be a compensatory mechanism to promote DOT1L activity and has been linked to ageing. Nevertheless, our data in the patient cartilage samples suggest that regulation of DOT1L activity is the main mechanism to control H3K79 methylation and its effects on gene transcription. The intrinsic catalytic activity of DOT1L is considered relatively low and the limited number of DOT1L

molecules in the cells does not match the high number of histones[36]. Thus, DOT1L should be directed to and activated at particular stretches of the DNA[36]. For instance, trans-histone cross-talk with ubiquitination at H2B not only interacts with DOT1L but contributes to the positioning of the enzyme to optimize H3K79 methylation[37]. Cumulative data suggest that there is no specific demethylase for H3K79 (ref. 36). Thus, demethylation of H3K79 appears to be largely due to histone renewal and cell division.

Our results identify a critical interaction between DOT1L and SIRT1 in articular chondrocytes. SIRT1 is a deacetylase with effects on epigenetic regulation of gene expression as well as other molecules, thus influencing different pathways[38]. We detected protein–protein interactions between DOT1L and SIRT1 in the transcriptional complex assembling upon activation of Wnt signalling. Within these complexes, DOT1L negatively regulates SIRT1 activity. Again, the downstream effects of the DOT1L-SIRT1 interaction appear to be strongly context-dependent. Whereas in chondrocytes inhibition of DOT1L results in increased Wnt signalling dependent on SIRT1, in DOT1L mediated mixed-lineage leukaemia, SIRT1 is part of an antitumoral repressive complex[39]. In the collecting ducts in the kidney, DOT1L and SIRT1 interact to suppress the expression of the epithelial Na(+) channel α-subunit (alpha-ENaC). Inhibition of SIRT1 resulted in higher levels of alpha-ENaC[40].

The deleterious effects of SIRT1 reported in this study may appear to be in contrast with its perceived role in cartilage biology and osteoarthritis[41]. Studies in genetic models have indicated that lack of SIRT1 activity in cartilage results in delayed growth and spontaneous osteoarthritis[42,43], as well as increases the severity of osteoarthritis in the destabilization of the medial meniscus model[44]. SIRT1 activator resveratrol protects against osteoarthritis in the same model[45]. However, in vitro, this drug triggers chondrocyte hypertrophy[46]. This apparent discrepancy with our observations can be explained by the broad biological effects of SIRT1, including control of metabolism and mitochondrial activity. The deacetylase activity of SIRT1 is not limited to histone modifications but also affects other molecules in the cell, including transcription factors such as forkhead proteins[47]. In addition, in the absence of SIRT1, DOT1L activity and the composition of these multi-protein complexes may be altered, thereby potentially affecting its protective role in cartilage.

The severe growth retardation in Dot1l[Cart-KO] mice remains intriguing. Further analysis suggests that the absence of Dot1l in chondrocytes disrupts the architecture of the growth plate with increased expression of COLX and MMP-13, markers of hypertrophic differentiation. Taking into account that we did not observe a role for DOT1L as a key regulator of Wnt signalling in primary osteoblasts, novel insights into the role of DOT1L in development and growth may result from experiments with other Cre-drivers such as Prx1. Obviously, specific attention should be given to growth retardation in children affected by leukaemia that are being treated with DOT1L inhibitors.

Further research will also be required to translationally validate the DOT1L/SIRT1 balance as a therapeutic target. Our different rescue and silencing experiments suggest specificity of the observed effects in vivo and in vitro. The severe growth phenotype in Dot1l[Cart-KO] mice precludes their use in induced or ageing models of osteoarthritis. Inducible conditional models, for instance using a tamoxifen or doxicyclin dependent collagen type II or aggrecan Cre-driver, may overcome these issues, although leakiness or postnatal activity loss of the Cre transgenes can be a limitation[48]. Nevertheless, such approaches will be necessary to further understand the role of DOT1L in joint disease, and more in particular in post-traumatic or

ageing-associated osteoarthritis. Moreover, the association between polymorphisms in DOT1L and osteoarthritis has been most strongly demonstrated for hip osteoarthritis[6]. Inducible tissue-specific genetic models may be useful to understand eventual differences between hip and knee disease.

In summary, our study provides novel information on DOT1L in mammalian gene regulation and identifies DOT1L as a key regulator of transcriptional programs essential for cartilage health. The balance between DOT1L and SIRT1 activity determines the activation of Wnt signalling in cartilage, with excessive pathway activity resulting in osteoarthritis. New epigenetics-based strategies, in particular those targeting the DOT1L network, could provide an innovative way for therapeutic modulation of Wnt signalling in joint disease and the development of effective treatments of osteoarthritis.

## Methods

**Materials.** The specific DOT1L inhibitor EPZ-5676 was obtained from Chemietek. Lithium chloride and EX527 were purchased from Sigma, XAV-939 from Selleck and SRT1720 from Calbiochem. HiPerFect and all siRNAs except DOT1L siRNA were purchased from Qiagen (Supplementary Table 2). DOT1L siRNA and lipofectamine RNAimax were purchased from Invitrogen. Recombinant human WNT3A protein was purchased from R&D Systems.

**Mice.** All experiments with mice were approved by the Ethics Committee for Animal Research (KU Leuven, Belgium). Wild-type male C57Bl/6 mice were purchased from Janvier (Le Genest St Isle, France). Dot1l transgenic mice (Dot1l[Tg]) were obtained from the Knockout Mouse Project (KOMP) (CSD29070) (ref. 49). Heterozygous mice were crossed to Gt(ROSA)26Sor[tm1(FLP1)Dym] mice (Jax 003946) for the removal of the Stop-cassette, to obtain Dot1l mice in which exon 2 is flanked by loxP sites (Dot1l[fl]). These mice were further bred to Tg (Col2a1.Cre)<1Bhr> mice (Jax 003554) to generate conditional cartilage-specific Dot1l knockout mice (Dot1l[fl/fl]; Col2-Cre[+/−] or Dot1l[Cart-KO]). Genotypes of animals were confirmed by PCR on genomic ear DNA.

**EPZ-5676, XAV-939 and EX527 intra-articular injections.** The concentration range for the in vivo administration of DOT1L inhibitor EPZ-5676 was estimated based on reported pharmacokinetics data in mouse[50]. From the range that we tested in a pilot study, we selected the minimum effective concentration that showed to inhibit DOT1L methylation (5 mg kg$^{-1}$). DOT1L inhibitor EPZ-5676, Wnt inhibitor XAV-939 (0.1 and 0.5 mg kg$^{-1}$) (ref. 51), SIRT1 inhibitor EX527 (1.25 mg kg$^{-1}$) (ref. 52) or vehicle was injected intra-articularly in the right knee of male 8-week-old wild-type C57Bl/6 mice four times, with an interval of 4 days. PBS was injected in the left knee.

**Histology.** Wild-type C57Bl/6 mice injected with EPZ-5676, XAV-939, EX527 or vehicle were killed 2 or 4 weeks after the first injection. Right and left knees from these mice were fixed overnight at 4 °C in 2% formaldehyde, decalcified for 3 weeks in 0.5 M EDTA pH 7.5, and embedded in paraffin. Dot1l[fl/fl]; Col2-Cre[−/−] (Cre-neg) and Dot1l[Cart-KO] male and female littermate mice were killed at the age of 4 weeks and right knees were fixed overnight at 4 °C in 2% formaldehyde, decalcified for 2 weeks in 0.5 M EDTA pH 7.5 and embedded in paraffin. Hematoxylin-Safranin O staining and immunohistochemistry were performed on 5 μm thick sections. Severity of disease was determined by histological scores on Hematoxylin-Safranin O stained sections throughout the knee by a blinded investigator (five sections at 100 μm distance). Both cartilage damage and synovial hyperplasia were assessed based on OARSI guidelines[53]. Depth of lesion (0–6) was scored on frontal knee sections. Lesion grades represent the following features; 0: surface and cartilage morphology intact, 1: small fibrillations without loss of cartilage, 2: vertical clefts below superficial layer and some loss of surface lamina, 3-6: vertical clefts/erosions to the calcified cartilage extending, 3: less than 25%, 4: 25–50%, 5: 50–75% and 6: more than 75%. The score represents the mean of the tibial scores at the medial and lateral side. For synovial hyperplasia, severity was determined on a 0–3 scale. Scoring was done by two independent readers, blinded to the genotype.

**Subchondral bone plate histomorphometry.** Histomorphometry was performed on Safranin O stained sections using an Olympus IX83 microscope and digital image analysis using Osteomeasure software, as previously described[54]. First, a box with a fixed width (800 μm) and variable height with the upper limit at the transition of calcified cartilage to subchondral bone and the lower limit at the transition submchondral bone to growth plate was created from the digital image (referred to as subchondral bone area). Next, a new box with the upper limit matched to the first box and the lower limit at the transition of the subchondral bone plate to trabecular bone was generated (referred to as subchondral bone plate

area). Subchondral bone area and subchondral bone plate area were calculated using the Osteomeasure software. To correct for section artifacts, we do not show the subchondral bone plate surface area *per se*, but express the subchondral bone plate thickness as a ratio of the subchondral bone plate area to the subchondral bone area.

**Immunohistochemistry.** Immunohistochemistry was performed on paraffin embedded EDTA decalcified mouse knee sections and on paraffin embedded sections of human cartilage explants from osteoarthritic patients and control trauma patients. Heat-induced epitope retrieval was performed using a Citrate-EDTA buffer (pH 6.2) for 10 min at 98 °C. Next, sections were treated with 3% $H_2O_2$/methanol for 10 min to inactivate endogenous peroxidase activity. Then, sections were blocked in normal goat serum for 30 min and incubated overnight at 4 °C with the primary antibodies against COLX (Abcam, ab58632; dilution 1:250 in EPZ-5676-injected mice and 1:400 in cartilage-specific *Dot1l* knockout mice), β-catenin (Abcam, ab6302; dilution 1:750), MMP-13 (Abcam, ab39012; dilution 1:100 in EPZ-5676-injected mice and 1:175 in cartilage-specific *Dot1l* knockout mice), TCF1 (Abcam, ab96777; dilution 1:100) or H3K79me2 (Abcam, ab3594; dilution 1:100). Rabbit IgG (Santa Cruz, sc-2027) was used as a negative control. In addition, the ABC-amplification technique (avidin-biotin complex) (Vectastain ABC kit, Vector Laboratories, USA) was used, except for the immunohistochemical detection of H3K79me2. Finally, peroxidase goat anti-rabbit IgG (Jackson Immunoresearch, Suffolk, UK) was applied for 30 min and peroxidase activity was determined using DAB. For the detection of COLX, antigen retrieval was performed enzymatically with 10 mg ml$^{-1}$ Hyaluronidase (Sigma-Aldrich, H3884) in $MgCl_2$- free PBS, for 40 min at 37 °C. In this case, endogenous peroxidase activity was blocked after the incubation with the primary antibody.

**Skeletal staining.** *Cre-neg* and *Dot1l*$^{Cart-KO}$ male and female littermate mice were killed at the age of 4 weeks, eviscerated and fixed for 5 days at room temperature in 95% ethanol. After dissolving the fat in acetone for 1 day, cartilage was stained with Alcian Blue (8GX, Sigma-Aldrich; 20% glacial acetic acid, 80% ethanol and 150 mg l$^{-1}$ Alcian Blue) for 1.5 days. Skeletons were rinsed twice with 95% ethanol for 2 days, before being cleared in 1% KOH for 1 day. Mineralized tissue, including bone, was then stained with Alizarin Red (Sigma-Aldrich; 50 mg l$^{-1}$ in 2% KOH) for 1.5 days. Afterwards, skeletons were cleared in 2% KOH for 6–8 days and then moved to a solution of 2% KOH and glycerol (80:20, respectively) for 4 days. Finally, skeletons were passed every 24 h to a solution of 2% KOH and glycerol with subsequent decreasing ratios of 2% KOH (60:40, 40:60 and 20:80, respectively).

**Cell culture.** Human articular chondrocytes were isolated from the hips of patients undergoing total hip replacement surgery. The University Hospitals Leuven Ethics Committee and Biobank Committee approved the study and specimens were taken with patients' written consent. Healthy articular chondrocytes were obtained from patients undergoing hip replacement for osteoporotic or malignancy-associated fractures. In specimens from osteoarthritic patients, obtained during prosthesis surgery, cartilage tissue was first classified macroscopically as either intact or damaged as described previously[55] taking into account colour, surface integrity and tactile impression tested with a scalpel. Cartilage was dissected from the joint explant surfaces and then rinsed with saline. The tissue was cut into small pieces, using a sterile surgical blade. Cartilage explants were incubated with 2 mg ml$^{-1}$ pronase solution (Roche) for 90 min at 37 °C and digested overnight at 37 °C in 1.5 mg ml$^{-1}$ collagenase B solution (Roche) under continuous agitation. The preparation was filtered through a 70 μM strainer and cells were plated in culture flasks and cultured in a humidified atmosphere at 37 °C, 5% $CO_2$. After reaching confluency, cells were passaged 1:3. Culture medium consisted of DMEM/F12 (Gibco), 10% fetal bovine serum (FBS) (Gibco), 1% (vol/vol) antibiotic/antimycotic (Gibco) and 1% L-glutamine (Gibco).

Primary human osteoblasts were isolated from the hips of patients undergoing total hip replacement surgery, using a modification of Beresford's procedure[56]. Trabecular bone was removed mechanically from the femur head, washed several times with sterile PBS to remove adherent cells and cut into 2–4 mm$^2$ pieces. These explants were agitated in culture medium (1:1 mixture of DMEM and Ham's F12 medium (Gibco) containing 1% (vol/vol) antibiotic-antimycotic (Gibco), 10% FBS (Gibco) and 1% L-glutamine (Gibco)), placed in a culture flask and incubated at 37 °C in a humidified atmosphere of 95% air and 5% $CO_2$. To favour osteoblastic differentiation, the standard culture medium was supplemented with 100 μg ml$^{-1}$ ascorbic acid (Sigma) and 10 mM β-glycerophosphate (Sigma). Non-adherent cells were removed after 3 days and afterwards the culture medium was changed twice a week. Primary cultures were maintained for 10–15 days until confluency (passage 0) when adherent cells were enzymatically released with 0.04% trypsin-EDTA solution. The resultant cell suspension was subcultured with standard cultured medium.

For the *in vitro* studies, cells were treated with 3 μM EPZ-5676, 20 mM LiCl, 1 μM XAV-939, 1 μM SRT1720 and/or 10 μM EX527, unless otherwise indicated. Small interfering RNA transfection was performed with 50 nM siRNA and HiPerFect or lipofectamine RNAimax (for DOT1L siRNA) for 48 h according to the manufacturer's instructions.

**Chondrocyte dedifferentiation assay.** Freshly isolated healthy human articular chondrocytes were seeded in monolayer and expanded from passage 0 up to passage 5. The cells were treated with 3 μM EPZ-5676 or vehicle every 3–4 days, during the cell expansion and associated dedifferentiation process. Protein and RNA were isolated at each passage when cells reached confluency. Effective DOT1L inhibition by EPZ-5676 at each passage was confirmed by Western blot analysis of H3K79 methylation. For the readout of the experiment, genes of interest were selected from a chondrocyte dedifferentiation transcriptome analysis taking into account the gene's known role in chondrocyte biology. The gene expression of those selected genes was measured by qPCR.

**Cell lysis and western blotting.** Cells were homogenized in IP Lysis/Wash buffer (Thermo Fisher) supplemented with 5% (vol/vol) Protease Mixture Inhibitor (Sigma) and 1 mM phenylmethanesulfonyl (Sigma). After two homogenization cycles (7 s) with an ultrasonic cell disruptor (Microson; Misonix), total cell lysates were centrifuged 10 min at 18,000g, and supernatant containing proteins was collected. The protein concentration of the extracts was determined by Pierce BCA Protein Assay Kit (Thermo Scientific). Immunoblotting analyses were performed as described in previous studies[3]. Antibodies against Actin (Sigma, A2066; dilution 1:4,000), active b-catenin (Merck Millipore, 05-665, clone 8E7; dilution 1:1,000), total β-catenin (BD biosciences, 610154, clone 14/β-catenin; dilution 1:2,000), DOT1L (Bethyl, A300-954A; dilution 1:1,000), GAPDH (Ambion, AM4300, clone 6C5; dilution 1:10,000), total H3 (Abcam, ab1791; dilution 1:10,000), H3K79me2 (Abcam, ab3594; dilution 1:1,000), and SIRT1 (Cell signalling, 2496, clone C14H4; dilution 1:1,000) were used following manufacturer's instructions. The blotting signals were detected using the SuperSignalWest Femto Maximum Sensitivity Substrate system (Thermo Scientific).

**Quantitative PCR.** Total RNA was extracted using the Nucleospin RNA II kit (Macherey-Nagel). cDNA was synthesized using 500 ng RNA isolated from human articular chondrocytes with the RevertAidHminus First Strand cDNA synthesis kit (Fermentas) according to the manufacturers' recommendations. Quantitative PCR analyses were performed as described previously using Maxima SYBRgreen qPCR master mix system (Fermentas)[6]. Gene expressions were calculated following normalization to housekeeping gene S29 mRNA levels using the comparative Ct (cycle threshold) method. The following PCR conditions were used: incubation for 10 min at 95 °C followed by 40 amplification cycles of 15 s of denaturation at 95 °C followed by 45 s of annealing-elongation at 60 °C. Melting curve analysis and 1% agarose gel migration of amplicons were performed to determine the specificity of the PCR. Primers used for qPCR analysis are listed in Supplementary Table 3.

**Microarray hybridization and data acquisition.** For the microarray, human articular chondrocytes were obtained from five non-OA hip fracture patients. The cells were treated with 3 μM EPZ-5676 or vehicle control for 4 days. Each sample was divided in two: one half was used for protein extraction and the other half for RNA extraction. Effective DOT1L inhibition by EPZ-5676 was confirmed by Western blot analysis of H3K79 methylation. RNA was isolated with Nucleospin RNA II kit (Macherey-Nagel). RNA concentration and purity were assessed with a NanoDrop Spectrophotometer (NanoDrop Technologies, Centreville, DE, USA) and integrity was determined using RNA nanochips and the Agilent 2100 Bio-analyzer (Agilent Technologies, Diegem, Belgium). Only non-degraded RNA without impurities (RNA integrity number > 7.7), was considered for microarray analysis. Transcriptional profiles were analysed by the VIB Microarray Facility (Nucleomics Core. Microarrays, nCounter, next-gen sequencing and bioinformatics. http://www.microarray.be). Per sample, 2 μg of total RNA spiked with bacterial RNA transcript positive controls (Affymetrix, Santa Clara, CA, USA) was converted to double stranded cDNA. Subsequently, the sample was converted and amplified to antisense cRNA and labelled with biotin. A mixture of purified and fragmented biotinylated cRNA and hybridization controls (Affymetrix) was hybridized on Affymetrix Human U133 Plus 2.0 arrays followed by staining and washing in a GeneChip fluidics station 450 (Affymetrix). To assess the raw probe signal intensities, chips were scanned using a GeneChip scanner 3000 (Affymetrix). The differentially expressed genes were explored with the limma package. Genes with nominal *P* values < 0.05 were selected for the enrichment analysis. DAVID version 6.7 (http://david.abcc.ncifcrf.gov/)[57] or Toppgenesuite (https://toppgene.cchmc.org)[58] tools were used for gene network analyses including categories KEGG pathway, GO molecular function, GO biological process, human and mouse phenotypes. A minimum gene count of 12 representing 1% of the differentially regulated genes recognized by DAVID was applied in the analysis. PANTHER database version 11.1 (http://pantherdb.org)[59] was used for graphical representation of functional categories using the 'PANTHER GO-slim' and Protein Class ontology resources.

**Luciferase assay.** Primary human chondrocytes were treated with vehicle or 3 μM EPZ-5676 for 14 days. On day 15, cells were plated into 24-well plates and they were transfected when they were 60–70% confluent with mGFP control plasmid (Origene PS100040) and Super8X TOPFlash or Super8X FOPFlash (TOPFlash mutant control) (Addgene plasmids #12456 and #12457, respectively) for canonical Wnt signalling reporter using Lipofectamine LTX Reagent with PLUS Reagent

(Life Technologies) according to the manufacturer's protocol. After 24 h, cells were treated with 20 mM LiCl with or without 3 μM EPZ-5676 and collected 24 h later. Luciferase assay was performed using Promega Luciferase Assay system according to the manufacturer's protocol. Briefly, cells were lysed with 400 μl per well 1 × lysis buffer (E1531) and underwent two freeze-thaw cycles to ensure complete cell lysis. Afterwards, 20 μl of cell lysate was transferred to opaque 96-well plates, and 50 μl of Luciferase Assay Reagent was dispensed into the plate. Luciferase level was measured using Plate-Reading Luminoskan Ascent (Thermo Scientific).

**ChIP analysis.** Chromatin immunoprecipitation assays were carried out using the Agarose ChIP kit from Thermo Scientific, according to the manufacturer's guidelines. Briefly, cell samples were crosslinked by 1% formaldehyde for 10 min, and the reaction was stopped by the addition of glycine to a 125 mM final concentration. The fixed cells were lysed in SDS buffer, and the chromatin was fragmented by micrococcal nuclease digestion. The sheared chromatin was incubated with antibodies against DOT1L (Bethyl, A300-954A; dilution 1:50), GCN5 (Santa Cruz, sc-20698; dilution 1:20), H3K4me3 (Abcam, ab8580; dilution 1:100), H3K9ac (Abcam, Ab4441; dilution 1:125), H3K79me2 (Abcam, Ab3594; dilution 1:100), EP300 (Abcam, ab14984, clone 3G230/NM-11), PPARGC1A (Santa Cruz, sc-13067; dilution 1:20), SIRT1 (Abcam, Ab12193; dilution 1:100) and recovered by binding to protein A/G agarose. Eluted DNA fragments were used directly for qPCR. Primers used for ChIP-qPCR analysis are listed in Supplementary Table 4.

**Co-Immunoprecipitation.** Co-Immunoprecipitation experiments were performed using the Pierce Co-Immunoprecipitation Kit (Thermo Scientific). Columns were conditioned following the manufacturer's recommendations. Antibody binding to the column was performed using 75 μg of either a mock antibody (donkey anti-goat IgG) as a control or DOT1L antibody (R&D Systems, MAB6546, clone 653613). After antibody immobilization, the columns were washed, and 100 μg of the lysate's proteins were incubated overnight at 4 °C under constant mixing. After four washings, retained proteins were eluted using 40 μl of Elution Buffer (Thermo Fisher) pH 3, and stored at − 80 °C. Protein complexes were then detected by Western blotting.

**SIRT1 activity assay.** The deacetylase activity of SIRT1 was determined by using a SIRT1 Activity Assay Kit (Abcam, ab156065) as recommended by the manufacturer. Briefly, cells were washed with cold PBS, lysed in IP Lysis/Wash buffer (Thermo Fisher) and incubated on ice for 5 min. After two homogenization cycles (7 s) with an ultrasonic cell disruptor (Microson; Misonix), total cell lysates were centrifuged 10 min at 13,000g, and supernatant containing proteins was collected. The protein concentration of the extracts was determined by Pierce BCA Protein Assay Kit (Thermo Scientific). Cell lysates (200 μg) were incubated with anti-SIRT1 antibody (10 μg) (Abcam, ab7343) for 3 h at 4 °C and then with protein A Agarose beads for the next 1.5 h. Precipitates were incubated with Fluoro-Sub-strate Peptide Solution, NAD$^+$ and SIRT1 Assay Buffer. Fluorescence intensity was then measured using a microtitre plate fluorometer with excitation at 360 nm and emission at 460 nm.

**Statistical analysis.** Data are presented as mean and s.e.m. where indicated. No statistical method was used to predetermine sample size for the animal experiments as the initial intervention (EPZ-5676 treatment) was compared against a condition expected to be close to normal. Additional interventions served as recue for the EPZ-5676 effects. The experiments were not randomized. Statistical analyses were performed where appropriate with GraphPad Prism software. For two group analyses, Student's $t$-test was used. For group analysis, one-way ANOVA with post hoc tests taking into account multiple comparisons were used. For repeated measurements two-way ANOVA was used analysing the effect of time, intervention and the interaction between time and intervention. If the interaction $P$ value was smaller than 0.05, post hoc tests were performed as above as appropriate. All tests were two-tailed.

**Data availability.** Microarray data that support the findings of this study have been deposited in Gene Expression Omnibus with the primary accession code GSE77916. The authors declare that all other data supporting the findings of this study are available within the article and its supplementary files, or available from the authors upon request.

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

## Acknowledgements

We are grateful to L. Storms for her technical support in this study and for, together with A. Hens, taking care of the animal facility management. We also thank F. Luyten, P. Tylzanowski and A. Liston for critically reading our manuscript, and Naomi Dirckx for assistance with digital image analysis. We are indepted to the traumatology and orthopedic surgeons willing to contribute samples (A. Sermon, J.P. Simon and S. Nys) as well as the nursing staff of the surgical theater (in particular M. Penninckx) This work was supported by grants from the Flanders Research Foundation (FWO-Vlaanderen), by DevRepair - Belspo IAPVII-07 and by Marie-Curie Intra-European postdoctoral fellowships to S.M. and F.C.

## Author contributions

R.J.L., F.C. and S.M. initiated the study. S.M, F., M.F.C., F.C., P.C. and R.J.L. planned the study and designed the experiments. S.M., F.M.F.C., C.A.-L., P.Y. and L.-A.G. performed the experiments. S.M, F.C, F.M.F.C, P.C and R.J.L wrote the manuscript.

## Additional information

**Competing interests:** The authors declare no competing financial interests.

