## [Peer Review File · Nature Communications]

Reviewers' comments:

Reviewer #1 (Remarks to the Author):

This is an exciting study providing insights into the function of the epigenetic regulator Dot1l in cartilage and osteoarthritis. This is important as Dot1l is one of the few genes linked to OA through human GWAS studies, but its molecular role in the joint has been relatively unclear. This study provides a thorough and quite convincing analyses of this role, using a combination of in vitro and in vivo, and mouse and human, approaches. However, several concerns need to be addressed.

1) The link between Dot1l and Wnt signaling in chondrocytes has been shown in the original 2012 paper (PNAS) implicating this gene in OA. This is not completely novel.

2) While the overall amount of data is impressive, the in vivo studies could be expanded – they are usually done at one time point only, with one concentration of EPZ, with very cartilage-centric outcomes

3) The authors paint a relatively straight pathway from Dot1l through Sirt1 to Wnt. The reality is probably more complex as all these components interact with many other pathways. For example, LiCl is used as Wnt antagonist – this is true, but as a GSK-3 inhibitor it will also activate other pathways such as HH and insulin/IGF. This should at least be discussed, or selected experiments be repeated with an actual canonical Wnt protein.

4) Several other groups have reported a chondro-protective role of Sirt1 which is in contrast to the results shown here. Context-specific roles of this protein appear plausible but this needs to be discussed thoroughly.

Additional points:

What is the source of the human healthy articular chondrocytes? OA chondrocytes were from hip replacements, but the origin of the healthy chondrocytes is unclear.

Fig. 1D: EPZ-injected joints appear to have much more subchondral bone. Is that seen reproducibly or just in this sample?

Fig. 2G: the panels need to be labelled, it is not clear which panels have which treatments. Overall none of these panels show clear OA, even in comparison to 1D? The vehicle should be included in the histological image.

The developmental phenotype of the Dot1l KO mice is fascinating. Some more characterization of the phenotype would greatly strengthen this manuscript. For example, is chondrogenesis delayed, which would fit with the data presented (although another Cre driver such as Prx or Dermo1 would be better to examine early chondrogenesis).

Is hypertrophy affected (which of course is highly relevant to OA) ? Canonical Wnt signaling is a driver of hypertrophy so this would be interesting. It might also be important to examine hypertrophic markers (e.g. Collagen X) in their EPZ-injected joints.

The authors should also at least point out the need to generate inducible cartilage-specific Dot1l KO mice for OA studies.

The analyses of the microarrays data is relatively short and could be expanded.

Reviewer #2 (Remarks to the Author):

Monteagudo and coworkers uncover DOT1L-dependent histone methylation is a novel epigenetic pathomechanism in osteoarthritis (OA). They present first in vitro and in vivo evidence that DOT1L protects articular chondrocytes from dedifferentiation by SIRT1-dependent inhibition of canonical WNT signaling and demonstrate that inhibition of DOT1L is sufficient to induce osteoarthritis. The proposed concept could be very relevant for the pathogenesis of OA, although further studies in human samples would be helpful to further support the concept.

Specific comments:

1. I may have missed it, but I did not find any expression analyses of DOT1L in human OA and experimental OA. Are the levels of DOT1L altered? If so, do they correlate inversely with histological disease scores? Do the levels of H3K79 correlated with severity? What drives the proposed downregulation of DOT1L in OA? In addition to damaged and non-damaged areas in samples from patients undergoing joint replacement secondary to OA, the authors should also provide data on completely unaffected cartilage, e.g. from trauma patients.
2. In most of their work on DOT1L and SIRT1, the authors exclusively work with small molecule inhibitors. To confirm those findings and to exclude off-target effects, key-findings should be replicated by genetic approaches.
3. The authors demonstrate that pharmacologic inactivation of DOT1L by intraarticular injections of EPZ-5676 induces OA-like changes. Those findings are very exciting. However, they also raise the question whether overexpression of DOT1L may protect from experimental OA. Moreover, does inhibition of DOT1L exacerbate experimental OA?
4. The authors show by IHC that inhibition of DOT1L induces accumulation of TCF1. To further strengthen the link between DOT1L and WNT signaling in vivo, the authors should also costain for nuclear beta-catenin (a more common readout) and DOT1L. Confirmation by WB would also be helpful.
5. Inhibition of DOT1L in chondrocytes induces the expression of the WNT-target genes LEF1, TCF1 and cMYC. However, the authors demonstrate that H3K79 and DOT1L only accumulate at the promoters of LEF1 and TCF1, but not of cMYC. How does DOT1L regulate cMYC?
6. The authors present interesting data in the cartilage-specific deletion of DOT1L by a constitutively active Col2Cre-deleter line. However, for OA, which develops at older age, an inducible deletion of DOT1L would be more relevant. This should at least be discussed.

Reply to the reviewers

We would like to thank the reviewers for their positive evaluation of our manuscript and the suggestions made. Their assessment helped us to substantially improve the manuscript. Detailed responses to the questions and comments are given below. We have highlighted the altered parts of the text in light gray in the revised manuscript.

Reviewer #1 (Remarks to the Author):

“This is an exciting study providing insights into the function of the epigenetic regulator Dot1l in cartilage and osteoarthritis. This is important as Dot1l is one of the few genes linked to OA through human GWAS studies, but its molecular role in the joint has been relatively unclear. This study provides a thorough and quite convincing analyses of this role, using a combination of *in vitro* and *in vivo*, and mouse and human, approaches. However, several concerns need to be addressed.”

We thank the reviewer for emphasizing the importance of our study and identifying that our work provides a thorough and quite convincing analysis of DOT1L’s role in osteoarthritis.

“1) The link between Dot1l and Wnt signaling in chondrocytes has been shown in the original 2012 paper (PNAS) implicating this gene in OA. This is not completely novel.”

The reviewer is correct in indicating that **our PNAS paper suggested interactions between Wnt signaling and DOT1L**. However, these experiments were performed with a mouse chondroprogenitor cell line (ATDC5 cells) and the effects observed in these precursor cells clearly differ from those observed in primary human articular chondrocytes. Previous reports also linked DOT1L with Wnt signaling in leukemia and in the intestine. Remarkably, these studies associate the activity of DOT1L and H3K79 methylation with increased transcription and active Wnt signaling. In contrast, DOT1L limits the activity of the Wnt cascade in primary human articular chondrocytes and *in vivo* in the mouse joint. To make this clear, **we now discuss the earlier work and the remarkable specificity of our findings**, also compared to primary human osteoblasts (Supplementary Fig. 8), **as part of the discussion in the revised manuscript**.

“In genome wide association studies, polymorphisms in the *DOT1L* gene were linked with cartilage thickness and osteoarthritis¹⁻³, as well as with height⁴. We earlier demonstrated that silencing of *Dot1l* in mouse chondroprogenitor cells negatively influences chondrogenic differentiation and that DOT1L interacts with Wnt pathway transcription factor TCF4 in these cells¹. An interaction between Wnt signaling molecule beta-catenin and DOT1L was further demonstrated in human cell lines^{5, 6}, in mouse intestinal crypts⁶, in colorectal cancer⁶, in drosophila⁵ and in zebrafish⁶. Remarkably, and in sharp contrast with our observations in the articular chondrocytes, DOT1L and H3K79 methylation have been mostly linked to increased Wnt target gene expression^{1, 5, 6}. Our observation that DOT1L prevents the deleterious hyper-activation of Wnt signaling in articular chondrocytes highlights that the regulatory effects of DOT1L on gene expression are strongly tissue and context dependent⁷, thereby opening opportunities for highly specific interventions.”

“2) While the overall amount of data is impressive, the *in vivo* studies could be expanded – they are usually done at one time point only, with one concentration of EPZ, with very cartilage-centric outcomes.”

The reviewer correctly pointed out that the *in vivo* experiments are reported at one time-point only and with one concentration of EPZ-5676.

However, before starting the *in vivo* experiments, **we first characterized the time-course of DOT1L inhibition and its downstream effects *in vitro*** in primary human articular chondrocytes. These experiments taught us that at least 4 days are required for inhibition of H3K79 methylation, and at least 2 weeks for biological effects. Our observations are in **agreement with available literature⁸**.

Indeed, the **concentration range for the *in vivo* administration** of DOT1L inhibitor EPZ-5676 was **estimated using reported pharmacokinetics data of EPZ-5676 in mouse⁸**. From the range that we tested in a pilot study and that is **shown in Fig. 1c**, we selected the **minimum concentration that was effective** to inhibit DOT1L methylation (5 mg/kg).

Although in our original submission we only reported data at one endpoint, this **first set of *in vivo* experiments** to induce osteoarthritis by inhibition of DOT1L **also included groups analyzed after only two weeks**. Already at two weeks, but to a lesser extent than at the later time point, EPZ-5676 injected mice showed increased signs of osteoarthritis compared to controls. **These data are now included in revised Fig. 1e of the revised manuscript**.

We also acknowledge the comment of the reviewer on the cartilage-centric outcomes reported in the original version of our manuscript as we **strongly agree that osteoarthritis is truly a disease of the whole joint organ**. In the original version, we already mentioned that we did not see any differences in the synovium. As suggested by the reviewer, we analysed the subchondral bone in these experiments using a digital image analysis that we developed earlier⁹. As for synovitis, no quantitative differences were seen between EPZ-5676 injected mice and controls. **We now include the quantitative assessment of synovitis and digital image analysis of the subchondral bone as new Supplementary Fig. 2**. We have also modified the text accordingly.

“We then studied whether loss of DOT1L activity *in vivo* triggers osteoarthritis, by intra-articular injection of EPZ-5676 into the knees of adult mice. H3K79 methylation was effectively inhibited by EPZ-5676 in articular chondrocytes (Fig. 1c). We observed increased cartilage damage by histology **2 and 4 weeks** after EPZ-5676 injections (Fig. 1d,e). **There were no differences between the groups in severity of synovitis or extent of subchondral bone remodeling, tissues that are known to show other features of osteoarthritis¹⁰** (Supplementary Fig. 2). Thus, these data suggest that DOT1L preserves articular cartilage homeostasis, protecting it against osteoarthritis.”

“3) The authors paint a relatively straight pathway from *Dot1l* through *Sirt1* to *Wnt*. The reality is probably more complex as all these components interact with many other pathways. For example, LiCl is used as *Wnt* antagonist – this is true, but as a GSK-3 inhibitor it will also activate other pathways such as HH and insulin/IGF. This should at least be discussed, or selected experiments be repeated with an actual canonical *Wnt* protein.”

We fully understand the reviewer's concern as the chondrocyte represents a complex biological system in which different pathways form networks with many interactions. Reviewer 2 expressed a related concern. We have chosen to use LiCl as a Wnt agonist (GSK3b-inhibitor) to **optimize the reproducibility of our experiments, for which we have consistently used primary human articular chondrocytes**. Primary cells are not continuously available and their *in vitro* expansion is limited. We took into account that the production of recombinant Wnt proteins still represents a challenge and **in our experience the bio-activity of commercial products is not always guaranteed, with batch to batch variability**.

However, **for the revision of this manuscript, as requested, key experiments were repeated with recombinant Wnt3a**. These experiments confirmed our key findings and are presented in **Supplementary Fig. 5a** (expression of Wnt target genes with and without DOT1L inhibition) and **Supplementary Fig. 5b** (ChIP-qPCR experiments to demonstrate that DOT1L and methylated H3K79 bind to Wnt target genes). In addition, the use of Wnt inhibitor XAV-939 in the *in vitro* and *in vivo* rescue experiments further confirms the specificity of the findings.

“4) Several other groups have reported a chondro-protective role of Sirt1 which is in contrast to the results shown here. Context-specific roles of this protein appear plausible but this needs to be discussed thoroughly.”

We agree with the reviewer that lack or inhibition of SIRT1 has been associated with a deleterious outcome in osteoarthritis models. SIRT1 is a complex regulator of different biological processes; therefore, this contrast with the effects of SIRT1 in the absence of DOT1L is not completely surprising. As argued by the reviewer, SIRT1 likely has context-specific roles. **We have further elaborated on the role of SIRT1 in cartilage in the general discussion at the end of the manuscript**.

“Our results identify a critical interaction between DOT1L and SIRT1 in articular chondrocytes. SIRT1 is a deacetylase with effects on epigenetic regulation of gene expression as well as other molecules, thus influencing different pathways¹¹. We detected protein-protein interactions between DOT1L and SIRT1 in the transcriptional complex assembling upon activation of Wnt signalling. Within these complexes, DOT1L negatively regulates SIRT1 activity. Again, the downstream effects of the DOT1L-SIRT1 interaction appear to be strongly context-dependent. Whereas in chondrocytes inhibition of DOT1L results in increased Wnt signaling dependent on SIRT1, in DOT1L mediated mixed-lineage leukemia, SIRT1 is part of an anti-tumoral repressive complex¹². In the collecting ducts in the kidney, DOT1L and SIRT1 interact to suppress the expression of the epithelial Na(+) channel alpha-subunit (*alpha-ENaC*). Inhibition of SIRT1 resulted in higher levels of *alpha-ENaC*¹³.

The deleterious effects of SIRT1 reported in this study may appear to be in contrast with its perceived role in cartilage biology and osteoarthritis¹⁴. Studies in genetic models have indicated that lack of SIRT1 activity in cartilage results in delayed growth and spontaneous osteoarthritis^{15, 16}, as well as increases the severity of osteoarthritis in the destabilization of the medial meniscus model¹⁷. SIRT1 activator resveratrol protects against osteoarthritis in the same model¹⁸. However, *in vitro*, this drug triggers

chondrocyte hypertrophy¹⁹. This apparent discrepancy with our observations can be explained by the broad biological effects of SIRT1, including control of metabolism and mitochondrial activity. The deacetylase activity of SIRT1 is not limited to histone modifications but also affects other molecules in the cell, including transcription factors such as forkhead proteins²⁰. In addition, in the absence of SIRT1, DOT1L activity and the composition of these multi-protein complexes may be altered, thereby potentially affecting its protective role in cartilage.“

Additional points:

“What is the source of the human healthy articular chondrocytes? OA chondrocytes were from hip replacements, but the origin of the healthy chondrocytes is unclear.”

We thank the reviewer for pointing this out. Our healthy articular chondrocytes are obtained from the trauma surgeons. Patients were undergoing hip replacement for osteoporotic or malignancy-associated fractures. We have modified the “Results” and “Methods” section to make this point clear.

“To study whether cartilage degeneration in osteoarthritis is related to changes in DOT1L activity, we performed immunohistochemistry of DOT1L-methylated H3K79 on **cartilage from non-osteoarthritic trauma patients and on preserved and damaged regions of cartilage from patients with osteoarthritis** (Fig. 1a).”

“Human articular chondrocytes were isolated from the hips of patients undergoing total hip replacement surgery. The University Hospitals Leuven Ethics Committee and Biobank Committee approved the study and specimens were taken with patients’ written consent. **Healthy articular chondrocytes were obtained from patients undergoing hip replacement for osteoporotic or malignancy-associated fractures. In specimens from osteoarthritic patients obtained during prosthesis surgery**, cartilage tissue was first classified macroscopically as either intact or damaged as described previously²¹ taking into account color, surface integrity and tactile impression tested with a scalpel.”

“Fig. 1D: EPZ-injected joints appear to have much more subchondral bone. Is that seen reproducibly or just in this sample?”

As indicated in our response to the comment that the analysis of the *in vivo* experiments was largely chondrocentric, we **have measured the subchondral bone** on histological sections using a method previously set-up in our laboratory and published in Osteoarthritis and Cartilage⁹. We did not find a difference between the groups. The results are presented in **new Supplementary Fig. 2a**.

“Fig. 2G: the panels need to be labelled, it is not clear which panels have which treatments. Overall none of these panels show clear OA, even in comparison to 1D? The vehicle should be included in the histological image.”

We **regret the lack of labeling in this figure which we have corrected**. Arrows have been added to indicate damaged areas. The data presented in Fig. 1 and revised Fig. 3 (formerly part of Fig. 2) come from different experiments in which biological variation is inevitable. The

severity of osteoarthritis in the models is limited which can be explained at least partially by the relative short time-window. However, this time-window was chosen taking into account the burden of repeated intra-articular injections. Nevertheless, **OARSI scores were consistent among blinded readers** as outlined in the “Methods” section and significantly different between groups.

“The developmental phenotype of the *Dot1l* KO mice is fascinating. Some more characterization of the phenotype would greatly strengthen this manuscript. For example, is chondrogenesis delayed, which would fit with the data presented (although another Cre driver such as *Prx* or *Dermo1* would be better to examine early chondrogenesis). Is hypertrophy affected (which of course is highly relevant to OA)? Canonical Wnt signaling is a driver of hypertrophy so this would be interesting. It might also be important to examine hypertrophic markers (e.g. Collagen X) in their EPZ-injected joints.”

We agree with the reviewer that the phenotype of the cartilage-specific *Dot1l* KO is intriguing. The use of other Cre drivers such as *Prx1* is certainly of interest although this may theoretically result in much more severe phenotypes taking into account the early embryonic lethality of full *Dot1l* KO mice.

We appreciated **the suggestion to further study markers of hypertrophy** in the context of this manuscript, as we indeed are aware that Wnt signaling is a strong driver of hypertrophy. While essential during development and growth, we are also in strong favor of the hypothesis that it is a key event that contributes to the development of osteoarthritis.

We have performed additional experiments addressing this question. As demonstrated in Fig. 5e, the **architecture of the growth plate** is clearly disturbed in *Dot1l^{Cart-KO}* mice. Both the pattern and the intensity of **COLX immunostaining** was different from controls in the growth plate and the articular cartilage of *Dot1l^{Cart-KO}* mice. We also looked at **MMP-13 staining**, which we found different from controls in the articular cartilage but not the growth plate of *Dot1l^{Cart-KO}* mice. **These data are presented in new Supplementary Fig. 8.**

As suggested, **we also evaluated markers of hypertrophy in mice treated with the EPZ-5676.** We found that both COLX and MMP-13 levels were increased in mice that received the intra-articular injections of EPZ-5676 compared to controls. **These data are presented in new Supplementary Fig. 4.**

The “results” section of the manuscript has been modified accordingly.

“Histology of the growth plates demonstrated a reduced and disorganized proliferative and pre-hypertrophic zone (Fig. 5e). Increased Wnt pathway activation in the absence of DOT1L was demonstrated by immunohistochemistry of TCF1 in the articular cartilage and growth plate of *Dot1l^{Cart-KO}* mice (Fig. 5f,g), with no detectable changes in beta-catenin activation (Supplementary Fig. 8a,b). **The pattern and the intensity of the immunoreactive signal for COLX was different in the growth plate of *Dot1l^{Cart-KO}* mice compared to controls (Supplementary Fig. 8d). Of note, COLX levels also appeared to be increased in the articular cartilage of *Dot1l^{Cart-KO}* mice (Supplementary Fig. 8c). Similarly, MMP-13 levels appeared to be increased in the articular cartilage of *Dot1l^{Cart-KO}* mice (Supplementary Fig. 8e) but not in the growth plate (Supplementary Fig. 8f).** Hence, these *in vivo* data demonstrate that DOT1L not

only regulates cartilage homeostasis but also skeletal growth, and caution against undesired growth effects when DOT1L inhibitors are used in children.”

“Chondrocytes are also found in the growth plate cartilage, a transient tissue that becomes gradually replaced by bone during skeletal development and growth. In developmental bone formation and in the growth plate, active Wnt signaling has an important role in terminal differentiation of these chondrocytes towards hypertrophic cells. These cells express type X collagen (COLX), upregulate matrix metalloproteinase-13 (MMP-13), and produce a calcified extra-cellular matrix. In osteoarthritis, hyperactivation of Wnt signaling is associated with ectopic hypertrophic differentiation^{22, 23}. Altered matrix composition and factors secreted by these hypertrophic-like articular chondrocytes, such as MMP-13, likely contribute to cartilage degeneration in osteoarthritis²³. **We detected increased immunohistochemical staining of COLX and MMP-13 in the articular cartilage of EPZ-5676-injected mice, particularly in the vicinity of lesions (Supplementary Fig. 4b,c).** Thus, DOT1L also protects against osteoarthritis by preventing Wnt-associated ectopic chondrocyte hypertrophy in the articular cartilage.”

Finally, we recapitulated these data and ideas in the discussion of this revised manuscript.

“The severe growth retardation in *Dot1l*^{Cart-KO} mice remains intriguing. Further analysis suggests that the absence of *Dot1l* in chondrocytes disrupts the architecture of the growth plate with increased expression of COLX and MMP-13, markers of hypertrophic differentiation. Taking into account that we did not observe a role for DOT1L as key regulator of Wnt signaling in primary osteoblasts, novel insights into the role of DOT1L in development and growth may result from experiments with other Cre-drivers such as *Prx1*. Obviously, specific attention should be given to growth retardation in children affected by leukemia that are being treated with DOT1L inhibitors.”

“The authors should also at least point out the need to generate inducible cartilage-specific *Dot1l* KO mice for OA studies.”

We fully agree with the reviewer that the use of inducible cartilage-specific *Dot1l* KO mice is of great interest, in particular to further understand the downstream effects and the potential for therapeutic intervention. As also outlined in our response to reviewer 2, such approaches, which are under way in our laboratory, are **not without challenges and the time-frame necessary is way beyond the commissioned time for revision**. In particular, the complex breeding, the duration of the different models that will be developed and the use of appropriate genetic controls is a time-consuming process that will take at least 1 to 2 years. Moreover, the use of inducible Cre models is associated with the risk of leakiness that affects the controlled deletion of the target and the effectiveness of Cre-deletion in postnatal cartilage has also been debated²⁴.

We therefore consider these approaches, including the study of osteoarthritis development in

surgical and aging models, as **the next steps in our quest to understand the role of DOT1L and Wnt signaling in this disease**. The planned experiments will serve as a basis to further tackle critical questions and to understand the downstream molecular impact of loss of DOT1L activity.

As suggested by the reviewer, we do aim to convey the importance of these next steps in the discussion.

“Further research will also be required to translationally validate the DOT1L/SIRT1 balance as a therapeutic target. Our different rescue and silencing experiments suggest specificity of the observed effects *in vivo* and *in vitro*. The severe growth phenotype in *Dot1l^{Cart-KO}* mice precludes their use in induced or ageing models of osteoarthritis. Inducible conditional models, for instance using a tamoxifen or doxycyclin dependent collagen type II or aggrecan Cre-driver, may overcome these issues, although leakiness or postnatal activity loss of the Cre transgenes can be a limitation²⁵. Nevertheless, such approaches will be necessary to further understand the role of DOT1L in joint disease, and more in particular in post-traumatic or ageing-associated osteoarthritis. Moreover, the association between polymorphisms in *DOT1L* and osteoarthritis has been most strongly demonstrated for hip osteoarthritis¹. Inducible tissue-specific genetic models may be useful to understand eventual differences between hip and knee disease.”

“The analyses of the microarrays data is relatively short and could be expanded.”

We agree with the reviewer that our manuscript did not contain a large and further exploratory analysis of the microarray data. **We would like to point out that the most important goal of our microarray experiment was to get a global view on the effects of DOT1L inhibition in primary human articular chondrocytes**. Therefore, we focused on the analyses that would stimulate our further hypothesis-driven approach and this is also what we report in the manuscript. The primary analysis was enrichment of well-annotated signaling pathways, in particular the **KEGG pathways**. **We also analysed a number of other systems that we have added as supplementary data (revised Supplementary Fig. 3)**. More specifically, we provide an overview of the type of molecules and biological processes that change after DOT1L inhibition and show enrichment for Gene Ontology term (molecular functions and biological processes), and for human and mouse phenotypes. **As highlighted in Supplementary Fig. 3, some of the enrichments fit strongly with a key role for DOT1L in joint and skeletal biology as a transcriptional regulator**.

We would like to point out to the reviewer that our original dataset is available in an open repository. This can allow the community, and in particular bioinformaticians, to further mine our dataset.

Reviewer #2 (Remarks to the Author):

“Monteagudo and coworkers uncover DOT1L-dependent histone methylation is a novel epigenetic pathomechanism in osteoarthritis (OA). They present first *in vitro* and *in vivo* evidence that DOT1L protects articular chondrocytes from dedifferentiation by SIRT1-dependent inhibition of canonical WNT signaling and demonstrate that inhibition of DOT1L is sufficient to induce osteoarthritis. The proposed concept could be very relevant for the

pathogenesis of OA, although further studies in human samples would be helpful to further support the concept.”

We would like to thank the reviewer for the assessment of our manuscript and the recognition that our proposed concept could be very relevant for the pathogenesis of osteoarthritis.

Specific comments:

“1. I may have missed it, but I did not find any expression analyses of DOT1L in human OA and experimental OA. Are the levels of DOT1L altered? If so, do they correlate inversely with histological disease scores? Do the levels of H3K79 correlated with severity? What drives the proposed downregulation of DOT1L in OA? In addition to damaged and non-damaged areas in samples from patients undergoing joint replacement secondary to OA, the authors should also provide data on completely unaffected cartilage, e.g. from trauma patients.”

We understand the comment of the reviewer. We therefore provide additional data in the manuscript. First, **we have compared the extent of H3K79 methylation not only in preserved and damaged cartilage from patients with osteoarthritis, but also in macroscopically normal articular cartilage from trauma patients.** As demonstrated in **revised Fig. 1a**, methylated H3K79 levels are high in normal and preserved cartilage, but low in damaged cartilage. Data presented in Fig. 1a are now representative of 5 osteoarthritis and 5 control patients.

In contrast, **we did not find differences in expression levels of *DOT1L*** using RNA from damaged and preserved areas of osteoarthritic cartilage and control cartilage. These experiments performed in our lab were confirmed by an analysis of the RAAK study RNAseq data (personal communication from I. Meulenbelt, Leiden University).

The manuscript was modified as follows:

“To study whether cartilage degeneration in osteoarthritis is related to changes in DOT1L activity, we performed immunohistochemistry of DOT1L-methylated H3K79 on **cartilage from non-osteoarthritic trauma patients and on preserved and damaged regions of cartilage from patients with osteoarthritis (Fig. 1a)**. This analysis revealed that the immunoreactive signal of methylated H3K79 was decreased **in damaged areas from osteoarthritic patients as compared to their corresponding preserved areas and to control cartilage. In contrast, *DOT1L* gene expression did not differ between damaged or preserved cartilage from patients with osteoarthritis, or control cartilage (data not shown)**. These observations suggest that DOT1L activity positively correlates with cartilage health.”

The search for factors that regulate the activity of DOT1L in cartilage has started in our laboratory as we fully agree with the reviewer that this is an intriguing question. This, in fact, represents a new project which we feel is beyond the scope of the current manuscript. We do discuss some potential regulatory mechanisms in the revised manuscript. Therefore the manuscript has been modified as follows:

“The factors that regulate DOT1L activity in the articular cartilage remain unknown and are an important area for further research. We did not detect differences in *DOT1L* gene expression levels between damaged or preserved cartilage from patients with

osteoarthritis, or control cartilage. However, this does not exclude transcriptional control as a mechanism to regulate DOT1L activity. In osteoarthritis, articular chondrocytes are not a uniform population and their healthy status may determine the expression of *DOT1L* at the individual cell level. Interestingly, some evidence suggests that pro-inflammatory signals activating NFκB signaling increase *DOT1L* expression²⁶. This may be a compensatory mechanism to promote DOT1L activity and has been linked to aging. Nevertheless, our data in patient cartilage samples suggest that regulation of DOT1L activity is the main mechanism to control H3K79 methylation and its effects on gene transcription. The intrinsic catalytic activity of DOT1L is considered relatively low and the limited number of DOT1L molecules in the cells does not match the high number of histones²⁷. Thus, DOT1L should be directed to and activated at particular stretches of the DNA²⁷. For instance, trans-histone cross-talk with ubiquitination at H2B not only interacts with DOT1L but contributes to the positioning of the enzyme to optimize H3K79 methylation²⁸. Cumulative data suggest that there is no specific demethylase for H3K79²⁷. Thus, demethylation of H3K79 appears to be largely due to histone renewal and cell division.”

“2. In most of their work on DOT1L and SIRT1, the authors exclusively work with small molecule inhibitors. To confirm those findings and to exclude off-target effects, key-findings should be replicated by genetic approaches.”

We fully understand the reviewer’s concern as the chondrocyte represents a complex biological system in which different pathways form networks with many interactions. Reviewer 1 expressed a related concern.

Key experiments initially performed with DOT1L inhibitor EPZ-5676 were replicated by siRNA-mediated silencing of the *DOT1L* gene in human articular chondrocytes. These novel data are now included in **revised Fig. 2e, 4a and 4f**.

We have chosen to use LiCl as a Wnt agonist (GSK3b-inhibitor) to **optimize the reproducibility of our experiments, for which we have consistently used primary human articular chondrocytes**. Primary cells are not continuously available and their *in vitro* expansion is limited. We took into account that the production of recombinant Wnt proteins still represents a challenge and **in our experience the bio-activity of commercial products is not always guaranteed, with batch to batch variability**.

However, **for the revision of this manuscript, as requested, key experiments were repeated with recombinant Wnt3a**. These experiments confirmed our key findings and are presented in **Supplementary Fig. 5a** (expression of Wnt target genes with and without DOT1L inhibition), **Supplementary Fig. 5b** (ChIP–qPCR experiments to demonstrate that DOT1L and methylated H3K79 bind to Wnt target genes). In addition, the use of Wnt inhibitor XAV-939 in the *in vitro* and *in vivo* rescue experiments further confirms the specificity of the findings.

“3. The authors demonstrate that pharmacologic inactivation of DOT1L by intraarticular injections of EPZ-5676 induces OA-like changes. Those findings are very exciting. However, they also raise the question whether overexpression of DOT1L may protect from experimental OA. Moreover, does inhibition of DOT1L exacerbate experimental OA?”

We appreciate the ideas and suggestions from the reviewer. As DOT1L activity does not appear to be strongly regulated at the gene expression level (see above), overexpression of the enzyme does not seem very useful. We are also interested in further confirming that DOT1L inhibition may exacerbate experimental OA. However, in this setting, we would need to repeatedly inject the DOT1L inhibitor intra-articularly over a 6 week period (e.g. in the DMM model). Such an intense scheme is likely to be damaging to the knees by itself and may raise concerns from our Ethical Committee for animal research. We are therefore currently exploring the use of inducible DOT1L KO mice (see below – reviewer question 6).

“4. The authors show by IHC that inhibition of DOT1L induces accumulation of TCF1. To further strengthen the link between DOT1L and WNT signaling in vivo, the authors should also co-stain for nuclear beta-catenin (a more common readout) and DOT1L. Confirmation by WB would also be helpful.”

The reviewer makes a **very interesting point** here. In fact, our manuscript already contained some of these data. In Fig. 2b, the blots (total protein lysate) show **that the amount of active beta-catenin increases upon LiCl stimulation of the chondrocytes but these increased levels do not appear to be influenced by EPZ-5676 treatment**. In contrast, upon EPZ-5676 treatment, the molecular interaction between DOT1L and beta-catenin is strongly decreased. Taking into account the suggestion of the reviewer, **we performed additional immunostaining for beta-catenin** in the articular cartilage of EPZ-5676 treated mice. Again, we did **not see differences in nuclear immunoreactivity suggesting that the effects of DOT1L take place downstream of the nuclear translocation of beta-catenin, as the ChIP-qPCR experiments further demonstrate**. This is shown in **new Supplementary Fig. 4**. We performed the same analyses in the *Dot1l^{Cart-KO}* mice, likewise not finding differences in immunoreactive signal as compared to controls. These data are shown in **new Supplementary Fig. 8**.

Our manuscript was revised accordingly:

“In Wnt reporter-transfected cells, EPZ-5676 increased the luciferase activity when the Wnt signaling pathway was activated by LiCl (Fig. 2c). **However, DOT1L inhibition did not induce detectable changes in active beta-catenin levels in human articular chondrocytes (Fig. 2b) or in the articular cartilage of EPZ-5676-injected mice (Supplementary Fig. 4a). This suggests that DOT1L regulates Wnt signaling downstream of beta-catenin stabilization.**”

“Increased Wnt pathway activation in the absence of DOT1L was demonstrated by immunohistochemistry of TCF1 in the articular cartilage and growth plate of *Dot1l^{Cart-KO}* mice (Fig. 5f,g), **with no detectable changes in beta-catenin activation (Supplementary Fig. 8a,b).**”

“5. Inhibition of DOT1L in chondrocytes induces the expression of the WNT-target genes LEF1, TCF1 and cMYC. However, the authors demonstrate that H3K79 and DOT1L only accumulate at the promoters of LEF1 and TCF1, but not of cMYC. How does DOT1L regulate cMYC?”

This is a good question. It is likely that *cMYC* expression is indirectly enhanced by other mechanisms different from DOT1L complex assembly at the gene promoters. Although the

underlying mechanism remains presently unknown, the absence of H3K79 methylation at the *cMYC* promotor represents an excellent negative control in the complex experiments.

“6. The authors present interesting data in the cartilage-specific deletion of DOT1L by a constitutively active Col2Cre-deleter line. However, for OA, which develops at older age, an inducible deletion of DOT1L would be more relevant. This should at least be discussed.”

As also outlined in our response to reviewer 1, **the use of inducible cartilage-specific Dot11 KO mice is of great interest**, in particular to further understand the downstream effects and the potential for therapeutic intervention. Such approaches, which are under way in our laboratory, are **not without challenges and the time-frame necessary is way beyond the commissioned time for revision**. In particular, the complex breeding, the duration of the different models that will be developed and the use of appropriate genetic controls is a time-consuming process that will take at least 1 to 2 years. Moreover, the use of inducible Cre models is associated with the risk of leakiness that affects the controlled deletion of the target and the effectiveness of Cre-deletion in postnatal cartilage has also been debated²⁴.

We therefore consider these approaches, including the study of osteoarthritis development in surgical and aging models, as **the next steps in our quest to understand the role of DOT1L and Wnt signaling in this disease**. The planned experiments will serve as a basis to further tackle critical questions and to understand the downstream molecular impact of loss of DOT1L activity.

As suggested by the reviewers, we do aim to convey the importance of these next steps in the discussion.

“Further research will also be required to translationally validate the DOT1L/SIRT1 balance as a therapeutic target. Our different rescue and silencing experiments suggest specificity of the observed effects *in vivo* and *in vitro*. The severe growth phenotype in *Dot11^{Cart-KO}* mice precludes their use in induced or ageing models of osteoarthritis. Inducible conditional models, for instance using a tamoxifen or doxycyclin dependent collagen type II or aggrecan Cre-driver, may overcome these issues, although leakiness or postnatal activity loss of the Cre transgenes can be a limitation²⁵. Nevertheless, such approaches will be necessary to further understand the role of DOT1L in joint disease, and more in particular in post-traumatic or ageing-associated osteoarthritis. Moreover, the association between polymorphisms in *DOT1L* and osteoarthritis has been most strongly demonstrated for hip osteoarthritis¹. Inducible tissue-specific genetic models may be useful to understand eventual differences between hip and knee disease.”

REFERENCES

1. Castaño Betancourt, M.C. et al. Genome-wide association and functional studies identify the DOT1L gene to be involved in cartilage thickness and hip osteoarthritis. *Proceedings of the National Academy of Sciences* **109**, 8218-8223 (2012).

2. Evangelou, E. et al. The DOT1L rs12982744 polymorphism is associated with osteoarthritis of the hip with genome-wide statistical significance in males. *Annals of the Rheumatic Diseases* **72**, 1264-1265 (2013).
3. Zhou, Y., Bi, F., Yang, G. & Chen, J. Association Between Single Nucleotide Polymorphisms of DOT1L Gene and Risk of Knee Osteoarthritis in a Chinese Han Population. *Cell Biochemistry and Biophysics* **70**, 1677-1682 (2014).
4. Lango Allen, H. et al. Hundreds of variants clustered in genomic loci and biological pathways affect human height. *Nature* **467**, 832-838 (2010).
5. Mahmoudi, T. et al. The Leukemia-Associated Mllt10/Af10-Dot1l Are Tcf4/ β -Catenin Coactivators Essential for Intestinal Homeostasis. *PLoS Biol* **8**, e1000539 (2010).
6. Mohan, M. et al. Linking H3K79 trimethylation to Wnt signaling through a novel Dot1-containing complex (DotCom). *Genes & Development* **24**, 574-589 (2010).
7. Gibbons, G.S., Owens, S.R., Fearon, E.R. & Nikolovska-Coleska, Z. Regulation of Wnt Signaling Target Gene Expression by the Histone Methyltransferase DOT1L. *ACS Chemical Biology* **10**, 109-114 (2015).
8. Basavapathruni, A. et al. Nonclinical pharmacokinetics and metabolism of EPZ-5676, a novel DOT1L histone methyltransferase inhibitor. *Biopharm Drug Dispos* **35**, 237-52 (2014).
9. Thysen, S., Luyten, F.P. & Lories, R.J. Loss of Frzb and Sfrp1 differentially affects joint homeostasis in instability-induced osteoarthritis. *Osteoarthritis Cartilage* **23**, 275-9 (2015).
10. Loeser, R.F., Goldring, S.R., Scanzello, C.R. & Goldring, M.B. Osteoarthritis: A disease of the joint as an organ. *Arthritis & Rheumatism* **64**, 1697-1707 (2012).
11. Feige, J.N. & Auwerx, J. Transcriptional targets of sirtuins in the coordination of mammalian physiology. *Current Opinion in Cell Biology* **20**, 303-309 (2008).
12. Chen, C.-W. et al. DOT1L inhibits SIRT1-mediated epigenetic silencing to maintain leukemic gene expression in MLL-rearranged leukemia. *Nat Med* **21**, 335-343 (2015).
13. Zhang, D., Li, S., Cruz, P. & Kone, B.C. Sirtuin 1 Functionally and Physically Interacts with Disruptor of Telomeric Silencing-1 to Regulate α -ENaC Transcription in Collecting Duct. *Journal of Biological Chemistry* **284**, 20917-20926 (2009).
14. Dvir-Ginzberg, M., Mobasher, A. & Kumar, A. The Role of Sirtuins in Cartilage Homeostasis and Osteoarthritis. *Current Rheumatology Reports* **18**, 43 (2016).
15. Gabay, O. et al. Sirtuin 1 enzymatic activity is required for cartilage homeostasis in vivo in a mouse model. *Arthritis & Rheumatism* **65**, 159-166 (2013).
16. Gabay, O. et al. Increased apoptotic chondrocytes in articular cartilage from adult heterozygous SirT1 mice. *Annals of the Rheumatic Diseases* **71**, 613-616 (2012).
17. Matsuzaki, T. et al. Disruption of Sirt1 in chondrocytes causes accelerated progression of osteoarthritis under mechanical stress and during ageing in mice. *Annals of the Rheumatic Diseases* **73**, 1397-1404 (2014).
18. Li, W., Cai, L., Zhang, Y., Cui, L. & Shen, G. Intra-articular resveratrol injection prevents osteoarthritis progression in a mouse model by activating SIRT1 and thereby silencing HIF-2 α . *Journal of Orthopaedic Research* **33**, 1061-1070 (2015).
19. Kim, H.J., Braun, H.J. & Dragoo, J.L. The effect of resveratrol on normal and osteoarthritic chondrocyte metabolism. *Bone and Joint Research* **3**, 51-59 (2014).
20. Motta, M.C. et al. Mammalian SIRT1 Represses Forkhead Transcription Factors. *Cell* **116**, 551-563 (2004).

21. Geyer, M. et al. Differential transcriptome analysis of intraarticular lesional vs intact cartilage reveals new candidate genes in osteoarthritis pathophysiology. *Osteoarthritis and Cartilage* **17**, 328-335 (2009).
22. Zhu, M. et al. Activation of β -Catenin Signaling in Articular Chondrocytes Leads to Osteoarthritis-Like Phenotype in Adult β -Catenin Conditional Activation Mice. *Journal of Bone and Mineral Research* **24**, 12-21 (2009).
23. Sun, M.M.-G. & Beier, F. Chondrocyte hypertrophy in skeletal development, growth, and disease. *Birth Defects Research Part C: Embryo Today: Reviews* **102**, 74-82 (2014).
24. Henry, S.P. et al. Generation of aggrecan-CreERT2 knockin mice for inducible Cre activity in adult cartilage. *Genesis* **47**, 805-14 (2009).
25. Henry, S.P. et al. Generation of aggrecan-CreERT2 knockin mice for inducible Cre activity in adult cartilage. *Genesis* **47**, 805-814 (2009).
26. Soria-Valles, C. et al. NF- κ B activation impairs somatic cell reprogramming in ageing. *Nat Cell Biol* **17**, 1004-1013 (2015).
27. Vlaming, H. & van Leeuwen, F. The upstreams and downstreams of H3K79 methylation by DOT1L. *Chromosoma* **125**, 593-605 (2016).
28. Vlaming, H. et al. Flexibility in crosstalk between H2B ubiquitination and H3 methylation in vivo. *EMBO reports* **15**, 1077-1084 (2014).

REVIEWERS' COMMENTS:

Reviewer #1 (Remarks to the Author):

The authors should be congratulated on an excellent and thorough response to my concerns. They provided a large amount of additional data that make the manuscript much stronger. I have no additional concerns.

Reviewer #2 (Remarks to the Author):

In general, the authors addressed the comments of the reviewers well. In some cases such as for the in vivo studies with additional CRE lines, additional experiments would have further increased the value of the study, but I agree that these experiments would have been rather challenging.